# Dendritic Localized Learning: Toward Biologically Plausible Algorithm

Changze Lv* [1]   Jingwen Xu* [1]   Yiyang Lu* [1]   Xiaohua Wang [1]   Zhenghua Wang [1]   Zhibo Xu [1]   Di Yu [2]
Xin Du [2]   Xiaoqing Zheng [1]   Xuanjing Huang [1]

## Abstract

Backpropagation is the foundational algorithm for training neural networks and a key driver of deep learning's success. However, its biological plausibility has been challenged due to three primary limitations: weight symmetry, reliance on global error signals, and the dual-phase nature of training, as highlighted by the existing literature. Although various alternative learning approaches have been proposed to address these issues, most either fail to satisfy all three criteria simultaneously or yield suboptimal results. Inspired by the dynamics and plasticity of pyramidal neurons, we propose Dendritic Localized Learning (DLL), a novel learning algorithm designed to overcome these challenges. Extensive empirical experiments demonstrate that DLL satisfies all three criteria of biological plausibility while achieving state-of-the-art performance among algorithms that meet these requirements. Furthermore, DLL exhibits strong generalization across a range of architectures, including MLPs, CNNs, and RNNs. These results, benchmarked against existing biologically plausible learning algorithms, offer valuable empirical insights for future research. We hope this study can inspire the development of new biologically plausible algorithms for training multilayer networks and advancing progress in both neuroscience and machine learning. Our code is available at https://github.com/Lvchangze/Dendritic-Localized-Learning.

## 1. Introduction

Backpropagation (Rumelhart et al., 1986) has been instrumental in the rapid development of deep learning (LeCun et al., 2015), establishing itself as the standard approach for training neural networks. Despite its undeniable success and widespread adoption in various applications ranging from image recognition (He et al., 2016; Dosovitskiy et al., 2020) to natural language processing (Devlin et al., 2019; Brown et al., 2020), the biological plausibility of backpropagation remains a subject of intense debate among researchers in both neuroscience and computational science (Bianchini et al., 1997; Payeur et al., 2021; Zahid et al., 2023).

The primary criticisms of backpropagation's biological plausibility stem from several unrealistic requirements: the symmetry of weight updates in the forward and backward passes (Stork, 1989), the computation of global errors that must be propagated backward through all layers (Crick, 1989), and the necessity of a dual-phase training process involving distinct forward and backward passes (Guerguiev et al., 2017; Hinton, 2022). These features lack clear analogs in neurobiological processes, which operate under the constraints of local information processing. Recognizing these limitations, the research community has made significant strides toward developing alternative training algorithms that could potentially align more closely with biological processes. Each of these approaches offers a unique perspective on how synaptic changes might occur in a biologically plausible manner, yet a consensus on an effective and biologically plausible training method remains out of reach.

In this paper, we first assess existing biologically plausible learning algorithms systematically, including feedback alignment (Lillicrap et al., 2016), local losses (Marblestone et al., 2016), predictive coding (Rao & Ballard, 1999; Whittington & Bogacz, 2017), perturbation learning (Williams, 1992; Werfel et al., 2003), target propagation (Bengio, 2014), Hebbian learning (Hebb, 1949; Munakata & Pfaffly, 2004), STDP (Song et al., 2000), the forward-forward algorithm (Hinton, 2022), and energy-based learning (Hopfield, 1984; Scellier & Bengio, 2017). From this review, we summarize three criteria that any learning algorithm must meet to be considered biologically plausible: **C1. Asymmetry of Forward and Backward Weights** – reflecting the inherent lack of symmetry in real synaptic connections; **C2. Local Error Representation** – ensuring computations are localized, without requiring a global error signal; **C3. Non-two-stage Training** – enabling the simultaneous occurrence of infer-

*Equal contribution [1]School of Computer Science, Fudan University [2]School of Software Technology, Zhejiang University. Correspondence to: Xiaoqing Zheng <zhengxq@fudan.edu.cn>.

*Proceedings of the 42nd International Conference on Machine Learning*, Vancouver, Canada. PMLR 267, 2025. Copyright 2025 by the author(s).

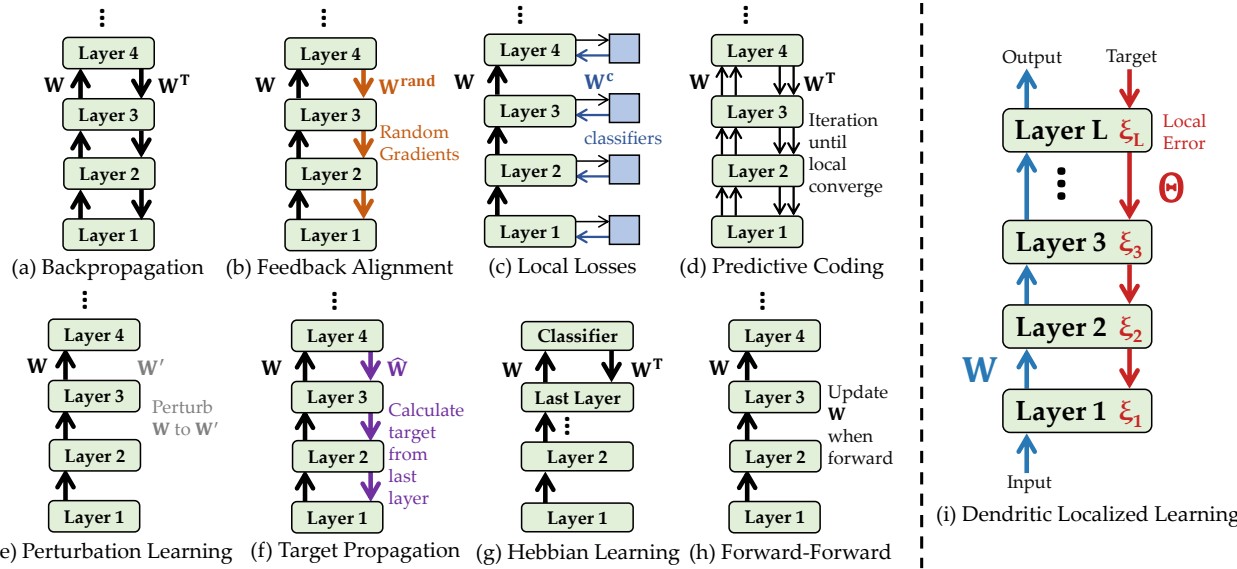

*Figure 1.* Illustrations of biologically plausible learning algorithms. (a) Backpropagation; (b) In feedback alignment, the weight matrix $\mathbf{W}$ is replaced with a random matrix during backpropagation; (c) In local losses, classic backpropagation is applied layer by layer; (d) In predictive coding, the transposed weights $\mathbf{W^T}$ are used iteratively for local convergence; (e) In perturbation learning, the weights $\mathbf{W}$ are randomly perturbed after the forward pass, generating a new $\mathbf{W'}$ for the next iteration; (f) In target propagation, two sets of weights are used: forward weights $\mathbf{W}$ and backward weights $\mathbf{\hat{W}}$, with $\mathbf{\hat{W}}$ used to calculate targets from the last layer; (g) In Hebbian learning, the final classification layer is trained using gradients; (h) The forward-forward algorithm updates weights during the forward pass; (i) In DLL, weights $\mathbf{W}$ and $\mathbf{\Theta}$ are asymmetric and updated simultaneously, with local errors being computed within the layer.

ence and training stages. We then conduct experiments to evaluate current algorithms across a variety of network architectures and real-world datasets to assess their performance and biological plausibility. We observe that algorithms that satisfy all three criteria tend to demonstrate significantly lower performance compared to backpropagation on benchmark datasets. What's worse, several algorithms may even fail to converge on specific architectures and tasks.

These limitations drive our exploration into the development of algorithms that both adhere to biological plausibility and maintain high performance. Inspired by pyramidal neurons (DeFelipe & Fariñas, 1992), which constitute approximately 70-85% of the total population of neurons in the cerebral cortex, we propose Dendritic Localized Learning (DLL), a novel learning algorithm that satisfies all three criteria and maintains strong performance. First, we model the pyramidal neuron as comprising three distinct compartments: the soma, apical dendrite, and basal dendrite. Evidence (Spruston, 2008) suggests that the apical dendrites of pyramidal neurons receive inputs from other cortical areas and non-specific thalamic sources, while the basal and side branches are primarily driven by inputs from lower-layer cells. Based on this, we propose that sensory input is directed to the basal dendrite, whereas the expected value is routed to the apical dendrite. The local error is then computed within the soma. Second, we propose the use of trainable backward weights to replace the transposed forward weights during

the backward pass, thereby ensuring compliance with the criterion of asymmetry weights. Lastly, in DLL, the information can be separated in space within a cell, then the two propagation phases, feedforward and feedback, do not require strict temporal segregation and hence could occur simultaneously. Through comprehensive experiments, we demonstrate the effectiveness of DLL on various benchmarks across diverse model architectures. Furthermore, we implement our DLL algorithm on time-varying recurrent neural networks (RNNs) and give a detailed derivation to support its theoretical rationality. Ultimately, we aim to not only highlight the current strengths and limitations of biologically plausible learning algorithms but also to stimulate further research and innovation in this vital area.

To conclude, our contributions can be summarized as:

- We review current biologically plausible learning algorithms and summarize 3 criteria for biological plausibility that an ideal learning algorithm should satisfy. We empirically benchmark these algorithms across diverse network architectures and datasets.
- We propose Dendritic Localized Learning (DLL), a learning algorithm satisfying all criteria of biological plausibility, to train multilayer neural networks.
- We conduct extensive experiments on leveraging the DLL algorithm to train MLPs, CNNs, and RNNs across various tasks, including image recognition, text char-

acter prediction, and time-series forecasting, showing comparable performance to backpropagation.

## 2. Criteria for Biological Plausibility

In this section, we first offer three criteria for biological plausibility that an ideal learning algorithm should satisfy, based on the methods reviewed. Secondly, we proceed to assess the current biologically plausible learning algorithms.

### 2.1. Criteria

Summarized from existing literature, we propose three criteria for evaluating the biological plausibility of learning algorithms:

**C1. Asymmetry of Forward and Backward Weights.** In conventional neural networks, forward-path neurons transmit their synaptic weights to the feedback path through a process known as weight transpose, which is considered biologically implausible. Real neurons are unlikely to share precise synaptic weights in this manner.

**C2. Local Error Representation.** Biological synapses are thought to adjust their strength based solely on local information, without relying on a global error signal. This is in stark contrast to chain-rule-based optimization methods, which typically compute error gradients using global information.

**C3. Non-two-stage Training.** Traditional training methods often involve distinct forward (inference) and backward (updating) phases, a feature absent in biological learning. However, the two propagation phases of a biological learning system, i.e., feedforward and backward, do not require strict temporal segregation and can occur simultaneously.

### 2.2. Assessment on Learning Algorithms

We show the detailed illustrations of the mentioned existing learning algorithms in Figure 1.

**Feedback Alignment** Feedback alignment (Lillicrap et al., 2016; Nøkland, 2016) modifies backpropagation by replacing the weight symmetry requirement with random feedback weights, which the network adapts to align with the true gradients over time. This approach demonstrates that perfect symmetry is not a strict requirement for learning. The gradient update is given by $\frac{\partial \mathcal{L}}{\partial \mathbf{W}} = \delta \cdot \mathbf{x}^T$, where $\delta = \mathbf{B} \cdot \frac{\partial \mathcal{L}}{\partial \mathbf{a}}$ represents the fixed random feedback weights. While this algorithm breaks the symmetry requirement of backpropagation, it still relies on global error signals and a two-stage training process.

**Local Losses** Local losses (Marblestone et al., 2016) method enables decentralized learning by assigning specific objectives to individual layers or regions of a network. The

local loss for layer $l$ is $\mathcal{L}^l = \|\mathbf{a}^l - \mathbf{t}^l\|^2$, where $\mathbf{a}^l$ is the activation, and $\mathbf{t}^l$ is a local target. Each layer minimizes its own loss independently: $\Delta \mathbf{W}^l = \eta \frac{\partial \mathcal{L}^l}{\partial \mathbf{W}^l}$. This method promotes scalability and modularity by avoiding reliance on global error signals.

**Predictive Coding** Predictive coding (Rao & Ballard, 1999; Whittington & Bogacz, 2017; Millidge et al., 2023; Salvatori et al., 2024) posits that the brain minimizes prediction error by iteratively updating its internal model to better predict incoming sensory inputs. This translates into hierarchical networks where each layer predicts the activity of the layer below. The loss function is $\mathcal{L} = \sum_l \|\mathbf{x}^{l-1} - \mathbf{u}^{l-1}\|^2$, where $\mathbf{x}^{l-1} = f^l(\mathbf{u}^l; (\mathbf{W}^l)^T)$ is the backpropagated activity of the lower layer. Weights are updated by backpropagating the prediction errors: $\Delta \mathbf{W}^l = \eta \frac{\partial \mathcal{L}}{\partial \mathbf{W}^l}$. Similar to local losses, although this method utilizes local error signals, it fails to satisfy **C1** and **C3**.

**Perturbation Learning** Perturbation learning (Williams, 1992) introduces random noise $\xi$ to the network's weights or inputs and observes the resulting change in output. The gradient is estimated using the finite difference method: $\Delta \mathbf{W} = \eta \frac{\Delta \mathcal{L}}{\xi}$, where $\Delta \mathcal{L}$ is the change in loss. If $\mathbf{W}$ are perturbed by $\mathbf{W} + \epsilon\xi$, the loss difference is: $\Delta \mathcal{L} \approx \mathcal{L}(\mathbf{W} + \epsilon\xi) - \mathcal{L}(\mathbf{W})$. However, the algorithm's reliance on distinct forward and perturbation phases indicates a departure from the simultaneous and seamless integration of inference and learning processes.

**Target Propagation** Target propagation (Bengio, 2014) addresses the biological implausibility of backpropagation by replacing error gradients with target activations. Each layer learns to approximate a target output that minimizes the overall error. The local objective for layer $l$ is $\mathcal{L}^l = \|\mathbf{a}^l - \mathbf{t}^l\|^2$, where $\mathbf{a}^l$ is the current activation, and $\mathbf{t}^l$ is the computed target. The weight update minimizes this local loss: $\mathbf{W}^l \leftarrow \mathbf{W}^l - \eta \frac{\partial \mathcal{L}^l}{\partial \mathbf{W}^l}$. Although vanilla target propagation satisfies all the criteria, it faces significant challenges in approximating the inverse function, leading to instability in convergence. Difference Target Propagation (Lee et al., 2014) is introduced to address this issue, but it violates **C3** because it requires the upper layers to propagate two separate values at different times to update the backward and forward weights.

**Hebbian Learning and STDP** Hebbian learning (Hebb, 1949) emphasizes strengthening the connection between neurons that frequently activate together. It forms the basis for synaptic plasticity in the brain. A more refined version, spike-timing-dependent plasticity (STDP) (Song et al., 2000), adjusts synaptic strengths based on the relative timing of pre and postsynaptic spikes. This mechanism allows networks to capture temporal correlations and has inspired algorithms for unsupervised learning and spiking neural

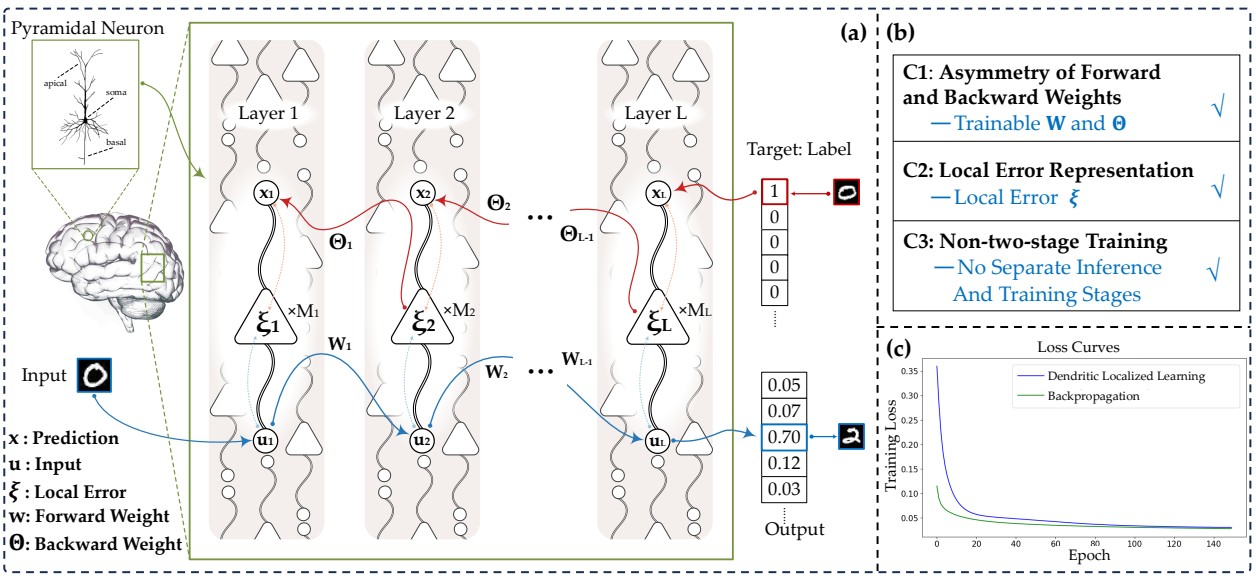

*Figure 2.* (a) Overview of Dendritic Localized Learning. (b) Our DLL algorithm satisfied all 3 criteria. (c) Models trained by DLL successfully converge and achieve comparable performance to those trained by backpropagation.

networks. The weight update in Hebbian learning is given by $\Delta w_{ij} = \eta x_i x_j$, where $x_i$ and $x_j$ are the activations of neurons $i$ and $j$. In STDP, the update depends on the spike timing difference

$$\Delta w_{ij} = \begin{cases} A^+ e^{-\Delta t/\tau^+}, & \Delta t > 0, \\ A^- e^{\Delta t/\tau^-}, & \Delta t \leq 0, \end{cases} \qquad (1)$$

where $A^+$, $A^-$ are scaling factors, and $\tau^+$, $\tau^-$ are time constants. These methods lack the ability to utilize supervised learning signals and coordinate weights across layers, with STDP further constrained by its reliance on precise spike timing.

**Forward-Forward**     Forward-forward learning (Hinton, 2022) eliminates the need for backward error propagation, training networks by separately optimizing for positive and negative samples in a forward-only manner. The layer-wise goodness function is defined as $g^l = \sum_i (a_i^l)^2$, where $a_i^l$ is the activation of neuron $i$ in layer $l$. The network maximizes $g^l$ for positive examples and minimizes it for negative examples: $\Delta \mathbf{W}^l = \eta \left( \frac{\partial g_{\text{pos}}^l}{\partial \mathbf{W}^l} - \frac{\partial g_{\text{neg}}^l}{\partial \mathbf{W}^l} \right)$. This method imposes significant constraints on input and processing, making it challenging to extend to other architectures such as CNNs.

**Energy-based Learning**     Energy-Based Learning defines an energy function $E(\mathbf{x}, \mathbf{y}; \mathbf{W})$, where $\mathbf{x}$ is the input, $\mathbf{y}$ is the output, and $\mathbf{W}$ are the model parameters. The goal is to minimize this energy such that desired outputs correspond to low-energy states. The weight updates are computed as $\frac{\partial \mathcal{L}}{\partial \mathbf{W}} = \frac{\partial E(\mathbf{x}, \mathbf{y}; \mathbf{W})}{\partial \mathbf{W}}$. Hopfield networks (Hopfield, 1984) use an energy function to model neural dynamics, where

the network evolves to stable states corresponding to stored patterns. Equilibrium Propagation (Scellier & Bengio, 2017) trains networks by reaching an equilibrium under inputs and then applying a small perturbation to the output. The loss minimized in this method may not align with the primary training objective, as reducing energy does not inherently lead to a corresponding decrease in task-specific loss.

## 3. Dendritic Localized Learning

In this section, we will introduce our proposed dendritic localized learning (DLL) and its detailed training procedure. Guided by three criteria proposed in Section 2.1, we draw inspiration from the pyramidal neuron (Spruston, 2008) and aim to simulate its calculation mode.

### 3.1. Three-Compartment Neurons

Pyramidal neurons are a type of excitatory neuron commonly found in the cerebral cortex (DeFelipe & Fariñas, 1992), playing a crucial role in processes like learning, memory, and higher cognitive functions. We follow Sacramento et al. (2018) to divide a pyramidal neuron into three compartments: soma, apical dendrite, and basal dendrite.

To satisfy criterion **C2**, i.e., local error representation, we define the local error of an individual neuron as $\xi = x - u$, where $u$ is the sensory input of the neuron and $x$ is the expected value, a.k.a, backpropagated activity, which is used to compute the error locally within the neuron. Under our setting, $\xi$ is calculated in the soma, and $u$ and $x$ are stored in the basal and apical dendrite, respectively. This division

aligns with the structure of pyramidal neurons, where different compartments are responsible for processing various types of information, contributing to the local error computation that facilitates learning. Therefore, for a neural network consisting of pyramidal neurons, the error of layer $i$ can be computed as:

$$\boldsymbol{\xi}_i = \mathbf{x}_i - \mathbf{u}_i, \tag{2}$$

where $\mathbf{x}_i$ is the backpropagated activity of layer $i + 1$.

However, due to criterion **C1** (asymmetry of forward and backward weights), calculating the backpropagated activity $\mathbf{x}_i$ from layer $i + 1$ becomes challenging, as the transpose of the forward weights $\mathbf{W}_{i+1}$ cannot be utilized. To address this, we introduce a special trainable weight $\boldsymbol{\Theta}$ to replace the traditional forward weight $\mathbf{W}$ during updating parameters.

Regarding criterion **C3** (non-two-stage training), we assume that information can be spatially separated within a neuron, allowing the two propagation phases to occur simultaneously without requiring strict temporal segregation. Additionally, we do not fix the forward weight $\mathbf{W}$, and instead, update both $\mathbf{W}$ and $\boldsymbol{\Theta}$ simultaneously, with supervision from the target label. The detailed training procedure will be discussed in Section 3.2.

Neurons leverage local information to dynamically update synaptic weights between adjacent layers, enabling the network to iteratively refine its architecture and functionality. Through this process, the output of each neuron gradually aligns with the intended target, facilitating an efficient redistribution of synaptic strength and embodying a biologically plausible learning mechanism.

To help readers better understand the mechanism of our DLL algorithm, we provide an illustration in Figure 2, assuming the training of an $L$-layer multilayer perceptron (MLP).

### 3.2. Training Procedure

DLL can be applied to various network architectures, including MLPs, CNNs, and RNNs. We use MLPs with MSE loss as an example to describe our method, then the total loss $\mathcal{L}$ of the whole neural network is:

$$\mathcal{L} = -\frac{1}{2} \sum_{i=1}^{L} \boldsymbol{\xi}_i^2 = -\frac{1}{2} \sum_{i=1}^{L} (\mathbf{x}_i - \mathbf{u}_i)^2, \tag{3}$$

where $\boldsymbol{\xi}_i$ is the loss of all neurons in layer $i$, and $L$ is the number of layers.

At the first epoch, for all layers except the last layer, we will initialize the expected value as the sensory input, written as:

$$\mathbf{u}_{i+1} = f(\mathbf{W}_i \mathbf{u}_i), \quad \mathbf{x}_{i+1} = \mathbf{u}_{i+1}, \tag{4}$$

where $f$ denotes the non-linear activation function, $\mathbf{u}_{i+1}$ is sensory input of layer $i + 1$, $\mathbf{x}_{i+1}$ is the expected value of

layer $i + 1$, and $\mathbf{W}_i$ is layer $i$'s weight. For the last layer, the $\mathbf{x}$ will be valued as the target label.

In the calculation of the backpropagated activity $\mathbf{x}$, we use the differentiation of loss $\mathcal{L}$ from $\mathbf{x}_i$ to obtain $\Delta \mathbf{x}_i$, which is the direction of change for $\mathbf{x}_i$. $\mathbf{x}_i$ depends solely on $\boldsymbol{\xi}_i$ and $\boldsymbol{\xi}_{i+1}$, as the parameter updates in the DLL algorithm are localized. Consequently, to compute $\Delta \mathbf{x}_i$, it is sufficient to use the derivatives of $\mathbf{x}_i$ with respect to $\boldsymbol{\xi}_i$ and $\boldsymbol{\xi}_{i+1}$ from $\mathcal{L}$. $\Delta \mathbf{x}_i$ is defined as:

$$\begin{aligned} \Delta \mathbf{x}_i &= \frac{\partial \mathcal{L}}{\partial \mathbf{x}_i} \\ &= \frac{\partial(-\frac{1}{2}\boldsymbol{\xi}_i^2 - \frac{1}{2}\boldsymbol{\xi}_{i+1}^2)}{\partial \mathbf{x}_i} \\ &= -\boldsymbol{\xi}_i + \frac{\partial \mathbf{u}_{i+1}}{\partial \mathbf{x}_i} \boldsymbol{\xi}_{i+1} \\ &= -\boldsymbol{\xi}_i + \mathbf{W}_i^T [\boldsymbol{\xi}_{i+1} \odot f'(\mathbf{W}_i \mathbf{u}_i)]. \end{aligned} \tag{5}$$

Here, $\odot$ denotes the Hadamard product. In our DLL algorithm, $\mathbf{x}$ propagates along the apical of the pyramidal neuron, specifically along the path defined by the parameters $\boldsymbol{\Theta}$, independent of the parameters $\mathbf{W}$ used in the forward pass. Therefore, the calculation formula for $\Delta \mathbf{x}_i$ is given by

$$\Delta \mathbf{x}_i = -\boldsymbol{\xi}_i + \boldsymbol{\Theta}_i^T [\boldsymbol{\xi}_{i+1} \odot f'(\mathbf{W}_i \mathbf{u}_i)]. \tag{6}$$

The updated value of $\mathbf{x}_i$ is calculated using the formula $\mathbf{x}_i \leftarrow \mathbf{x}_i + \eta_{\mathbf{x}} * \Delta \mathbf{x}_i$, where $\eta_{\mathbf{x}}$ denotes the learning rate of updating $\mathbf{x}$. Ultimately, when $\mathbf{x}_i$ approaches stability, $\Delta \mathbf{x}_i = 0$, leading to the expression

$$\boldsymbol{\xi}_i = \boldsymbol{\Theta}_i^T [\boldsymbol{\xi}_{i+1} \odot f'(\mathbf{W}_i \mathbf{u}_i)]. \tag{7}$$

Finally, we will adjust $\mathbf{W}_i$ through $\mathbf{W}_i \leftarrow \mathbf{W}_i + \eta_{\mathbf{W}} * \Delta \mathbf{W}_i$, where $\eta_{\mathbf{W}}$ is the learning rate for updating $\mathbf{W}$, and the update term $\Delta \mathbf{W}_i$ is:

$$\begin{aligned} \Delta \mathbf{W}_i &= \frac{\partial \mathcal{L}}{\partial \mathbf{W}_i} \\ &= \frac{\partial(-\frac{1}{2}\boldsymbol{\xi}_{i+1}^2)}{\partial \mathbf{W}_i} \\ &= -\boldsymbol{\xi}_{i+1}(-\frac{\partial \mathbf{u}_{i+1}}{\partial \mathbf{W}_i}) \\ &= \boldsymbol{\xi}_{i+1} \odot f'(\mathbf{W}_i \mathbf{u}_i)\mathbf{u}_i. \end{aligned} \tag{8}$$

For $\boldsymbol{\Theta}_i$, we will update it using the following rule: $\boldsymbol{\Theta}_i \leftarrow \boldsymbol{\Theta}_i + \eta_{\boldsymbol{\Theta}} * \Delta \boldsymbol{\Theta}_i$, where $\eta_{\boldsymbol{\Theta}}$ is the learning rate for updating $\boldsymbol{\Theta}$, and $\Delta \boldsymbol{\Theta}_i$ is the update term:

$$\begin{aligned} \Delta \boldsymbol{\Theta}_i^T &= \frac{\partial \mathcal{L}}{\partial \boldsymbol{\Theta}_i^T} \\ &= \frac{\partial \mathcal{L}}{\partial \boldsymbol{\xi}_i} \frac{\partial \boldsymbol{\xi}_i}{\partial \boldsymbol{\Theta}_i^T} \\ &= \frac{\partial \left(-\frac{1}{2}\boldsymbol{\xi}_i^2\right)}{\partial \boldsymbol{\xi}_i} \frac{\partial \boldsymbol{\Theta}_i^T [\boldsymbol{\xi}_{i+1} \odot f'(\mathbf{W}_i \mathbf{u}_i)]}{\partial \boldsymbol{\Theta}_i^T} \\ &= -\boldsymbol{\xi}_i [\boldsymbol{\xi}_{i+1} \odot f'(\mathbf{W}_i \mathbf{u}_i)]. \end{aligned} \tag{9}$$

*Table 1.* We show both the biological plausibility of various algorithms with the proposed criteria and the accuracy of image classification achieved by various bio-plausible learning algorithms. Our proposed DLL achieves the highest performance among the algorithms satisfying all criteria. "C1, C2, C3" stand for three criteria proposed in Section 2.1. All results are averaged across 4 random seeds.

| Method | C1 | C2 | C3 | Model | MNIST | FashionMNIST | SVHN | CIFAR-10 | Avg. |
|---|---|---|---|---|---|---|---|---|---|
| Backpropagation | ✗ | ✗ | ✗ | MLPs | $98.62\%_{\pm0.17\%}$ | $88.54\%_{\pm0.64\%}$ | $\mathbf{60.91\%}_{\pm0.42\%}$ | $48.74\%_{\pm0.56\%}$ | $\mathbf{74.20\%}$ |
| | | | | CNNs | $99.56\%_{\pm0.14\%}$ | $\mathbf{92.68\%}_{\pm0.42\%}$ | $\mathbf{95.35\%}_{\pm1.53\%}$ | $\mathbf{75.10\%}_{\pm0.54\%}$ | $\mathbf{90.67\%}$ |
| Feedback Alignment | ✓ | ✗ | ✗ | MLPs | $91.87\%_{\pm0.08\%}$ | $82.16\%_{\pm0.14\%}$ | $54.91\%_{\pm0.23\%}$ | $48.46\%_{\pm0.11\%}$ | $69.35\%$ |
| | | | | CNNs | $97.00\%_{\pm0.13\%}$ | $89.74\%_{\pm0.17\%}$ | $92.66\%_{\pm0.26\%}$ | $59.60\%_{\pm0.46\%}$ | $84.75\%$ |
| Local Losses | ✗ | ✓ | ✗ | MLPs | $98.56\%_{\pm0.19\%}$ | $88.07\%_{\pm0.38\%}$ | $59.12\%_{\pm0.27\%}$ | $48.58\%_{\pm0.35\%}$ | $73.58\%$ |
| | | | | CNNs | $99.39\%_{\pm0.06\%}$ | $91.90\%_{\pm0.26\%}$ | $95.08\%_{\pm0.25\%}$ | $72.18\%_{\pm0.10\%}$ | $89.64\%$ |
| Predictive Coding | ✗ | ✓ | ✗ | MLPs | $98.42\%_{\pm0.13\%}$ | $\mathbf{88.72\%}_{\pm0.65\%}$ | $59.05\%_{\pm0.45\%}$ | $47.34\%_{\pm0.24\%}$ | $73.38\%$ |
| | | | | CNNs | $99.41\%_{\pm0.40\%}$ | $92.03\%_{\pm0.70\%}$ | $94.53\%_{\pm1.54\%}$ | $72.94\%_{\pm0.32\%}$ | $89.72\%$ |
| Perturbation Learning | ✓ | ✓ | ✗ | MLPs | $91.44\%_{\pm0.40\%}$ | $68.90\%_{\pm0.47\%}$ | $48.15\%_{\pm1.06\%}$ | $31.07\%_{\pm0.31\%}$ | $59.89\%$ |
| | | | | CNNs | $92.61\%_{\pm0.43\%}$ | $75.79\%_{\pm0.83\%}$ | $57.69\%_{\pm1.32\%}$ | $39.72\%_{\pm0.38\%}$ | $66.45\%$ |
| Difference Target Propagation | ✓ | ✓ | ✗ | MLPs | $94.01\%_{\pm0.12\%}$ | $83.28\%_{\pm0.31\%}$ | $54.11\%_{\pm0.09\%}$ | $46.10\%_{\pm0.10\%}$ | $69.38\%$ |
| | | | | CNNs | $96.40\%_{\pm0.05\%}$ | $90.51\%_{\pm0.18\%}$ | $69.72\%_{\pm0.32\%}$ | $50.88\%_{\pm0.07\%}$ | $76.88\%$ |
| Hebbian Learning | ✓ | ✓ | ✓ | MLPs | $78.29\%_{\pm0.07\%}$ | $67.40\%_{\pm0.69\%}$ | $40.80\%_{\pm0.44\%}$ | $19.98\%_{\pm0.23\%}$ | $51.62\%$ |
| | | | | CNNs | $83.05\%_{\pm0.12\%}$ | $72.03\%_{\pm0.48\%}$ | $44.77\%_{\pm0.33\%}$ | $29.86\%_{\pm0.13\%}$ | $57.43\%$ |
| R-STDP | ✓ | ✓ | ✓ | MLPs | $77.18\%_{\pm0.17\%}$ | $70.03\%_{\pm0.28\%}$ | $41.76\%_{\pm0.46\%}$ | $22.68\%_{\pm0.30\%}$ | $52.91\%$ |
| | | | | CNNs | $91.67\%_{\pm0.04\%}$ | $74.29\%_{\pm0.30\%}$ | $50.02\%_{\pm0.32\%}$ | $33.19\%_{\pm0.38\%}$ | $62.29\%$ |
| Forward Forward | ✓ | ✓ | ✓ | MLPs | $96.99\%_{\pm0.14\%}$ | $80.51\%_{\pm0.74\%}$ | $47.52\%_{\pm0.63\%}$ | $39.48\%_{\pm0.10\%}$ | $66.12\%$ |
| | | | | CNNs | $15.66\%_{\pm0.08\%}$ | $10.00\%_{\pm0.00\%}$ | $6.70\%_{\pm0.00\%}$ | $10.32\%_{\pm0.00\%}$ | $10.67\%$ |
| Equilibrium Propagation | ✓ | ✓ | ✓ | MLPs | $93.81\%_{\pm0.18\%}$ | $75.65\%_{\pm0.35\%}$ | $22.62\%_{\pm0.39\%}$ | $16.93\%_{\pm0.10\%}$ | $52.25\%$ |
| | | | | CNNs | $26.73\%_{\pm0.22\%}$ | $30.26\%_{\pm0.29\%}$ | $6.70\%_{\pm0.00\%}$ | $10.32\%_{\pm0.00\%}$ | $18.50\%$ |
| DLL (Ours) | ✓ | ✓ | ✓ | MLPs | $97.57\%_{\pm0.40\%}$ | $87.50\%_{\pm0.43\%}$ | $\mathbf{56.60\%}_{\pm0.12\%}$ | $45.87\%_{\pm0.10\%}$ | $\mathbf{71.89\%}$ |
| | | | | CNNs | $98.87\%_{\pm0.30\%}$ | $90.88\%_{\pm0.40\%}$ | $85.81\%_{\pm0.17\%}$ | $70.89\%_{\pm0.58\%}$ | $86.61\%$ |

We present our global algorithm in Appendix A, which outlines the overall training procedure using DLL. Additionally, to enhance clarity, we provide a detailed derivation for training RNNs with DLL in Appendix B. This derivation highlights its application to sequential tasks and addresses the unique challenges introduced by temporal dependencies. The convergence properties and guarantees will be discussed in Appendix C.

# 4. Experiments

In this section, we first introduce the experimental settings, including datasets and implementation details. Secondly, we benchmark all biologically plausible learning algorithms, including our proposed DLL, on image classification tasks. Thirdly, we evaluate RNNs trained with DLL on text character prediction and time-series forecasting. Finally, we do an ablation study and analyze its inner properties.

## 4.1. Experimental Settings

**Image Classification**  We utilize several widely used benchmark datasets for image recognition tasks: MNIST, FashionMNIST, SVHN, and CIFAR-10. We take classification accuracy as the metric.

**Text Character Prediction**  We conduct next-character prediction with RNNs on Harry Potter (Rowling, 2019). We take Prediction Accuracy (Pred. Acc.) as the metric, which measures the proportion of correctly predicted characters among all predictions.

**Time-Series Forecasting**  We employ RNNs with various learning algorithms for real-world multivariate time-series forecasting, including Electricity (Lai et al., 2018), Metr-la (Li et al., 2017), and Pems-bay (Li et al., 2017).

To ensure fairness, we use the same model architecture across all learning algorithms for MLPs, CNNs, and RNNs under a certain dataset. Due to the difficulty of achieving convergence with vanilla STDP in this setting, we replace it with an improved version, reward-modulated STDP (R-STDP) (Mozafari et al., 2018). For detailed dataset statistics, metric explanations, and implementation of models, please refer to Appendix D.

## 4.2. Image Recognition

We benchmark all previous bio-plausible algorithms and our proposed DLL on image classification tasks in Table 1. As shown in Table 1, we can conclude that:

**Current biologically plausible algorithms, while offering valuable insights, often fall short of the high performance achieved by traditional backpropagation**, particularly on complex datasets like SVHN and CIFAR-10. Algorithms meeting one criterion, including feedback alignment, local losses, and predictive coding, show competitive performance on simple datasets like MNIST and Fashion-MNIST, indicating their potential viability. However, their reduced effectiveness on more challenging tasks, such as CIFAR-10, highlights the need for further advancements in this field. For paradigms that satisfy two criteria, we ob-

*Table 2.* Performance of RNNs trained with various learning algorithms on text-character prediction and time-series forecasting. The best results are formatted in **bold** font format. ↑ (↓) indicates the higher (lower) the better. All results are averaged across 3 random seeds.

| Method | Harry Potter | Electricity | | Metr-la | | Pems-bay | |
|---|---|---|---|---|---|---|---|
| | Pred. Acc. ↑ | MSE ↓ | MAE ↓ | MSE ↓ | MAE ↓ | MSE ↓ | MAE ↓ |
| Backpropagation | $\mathbf{51.9}\%_{\pm 1.0\%}$ | $0.175_{\pm 0.007}$ | $0.324_{\pm 0.007}$ | $\mathbf{0.131}_{\pm 0.004}$ | $\mathbf{0.214}_{\pm 0.005}$ | $\mathbf{0.164}_{\pm 0.001}$ | $\mathbf{0.190}_{\pm 0.002}$ |
| Predictive Coding | $38.8\%_{\pm 1.8\%}$ | $\mathbf{0.162}_{\pm 0.019}$ | $\mathbf{0.312}_{\pm 0.018}$ | $0.141_{\pm 0.001}$ | $0.228_{\pm 0.005}$ | $0.178_{\pm 0.004}$ | $0.202_{\pm 0.003}$ |
| DLL (Ours) | $33.7\%_{\pm 0.6\%}$ | $0.172_{\pm 0.018}$ | $0.321_{\pm 0.013}$ | $0.155_{\pm 0.005}$ | $0.264_{\pm 0.001}$ | $0.178_{\pm 0.005}$ | $0.224_{\pm 0.004}$ |

serve a noticeable performance reduction compared to those meeting only one criterion. Perturbation learning, which does not require a backward procedure, struggles to achieve adequate convergence.

**Our proposed DLL successfully integrates biological plausibility with high performance**. Among algorithms that meet all the criteria, MLPs or CNNs trained with DLL converge successfully across all datasets, even achieving performance comparable to backpropagation. This suggests that it is possible to reconcile biological plausibility with high performance. In contrast, methods like Hebbian learning, the forward-forward algorithm, and equilibrium propagation struggle to guarantee convergence, especially on more complex architectures (CNNs) or challenging datasets. For instance, MLPs trained with the forward-forward algorithm (Hinton, 2022) achieve comparable average accuracy while CNNs trained with that fail to converge across all benchmarks. In comparison, DLL consistently converges quickly and delivers relatively satisfactory results. The results of CIFAR-100 and Tiny-ImageNet are shown in Appendix E.

### 4.3. Sequential Tasks

Most biologically plausible learning paradigms mentioned in Section 2.2, such as local losses and target propagation, were primarily designed for discrimination tasks. Therefore, to assess the versatility of our proposed DLL, we follow Millidge et al. (2022) to conduct experiments on training RNNs with our DLL for sequential regression tasks, as shown in Table 2. We compare DLL against several algorithms enabling model convergence, including backpropagation and predictive coding. Methods that fail to converge are not included in the table.

First, we evaluate RNNs trained with different algorithms on a text-character prediction task to investigate their language processing abilities. Backpropagation outperforms other methods by a significant margin. While backpropagation performs best, predictive coding and DLL emerge as promising biologically plausible alternatives. Notably, DLL is the only method that satisfies all the biological plausibility criteria while still succeeding in converging during training. Furthermore, we regard real-world multi-variant time-series forecasting as an ideal regression task for evaluating a model's ability to capture temporal dependencies.

This task offers insights into DLL's ability to model the internal dynamics of sequential data. Table 2 illustrates that RNNs trained using DLL exhibit competitive performance across a range of tasks, achieving results on par with or surpassing those of backpropagation in several metrics, with particularly strong performance on the Electricity dataset. These results underscore DLL's effectiveness in capturing temporal dependencies while maintaining alignment with biologically plausible learning mechanisms.

What's more, we successfully employ our DLL method to train TextCNNs (Kim, 2014) for text classification tasks. We show the results on text classification in Appendix F.

### 4.4. Ablation Study

As mentioned in Section 3, we propose the weights $\Theta$ and its updating rules based on local errors. Taking the feedback alignment (FA) into consideration, we think it is necessary to evaluate how the convergence or performance will be influenced when $\Theta$ is a random and unchanged matrix. We name this special method as "DLL-FA", i.e., $\Theta$ will initialize randomly and not participate in the model updating procedure.

*Table 3.* Ablation experiments on $\Theta$. "DLL-FA" indicates $\Theta$ initializes randomly and will not be updated, combining our proposed DLL and feedback alignment (FA). Numbers with $^*$ indicate models fail to converge. ↑ (↓) indicates the higher (lower) the better.

| | Metric | DLL | DLL-FA |
|---|---|---|---|
| MLPs on MNIST | Acc. ↑ | $\mathbf{97.57}\%$ | $97.37\%$ |
| CNNs on CIFAR-10 | | $\mathbf{70.89}\%$ | $69.85\%$ |
| RNNs on Harry Potter | Acc. ↑ | $\mathbf{33.70}\%$ | $0.71\%^*$ |
| RNNs on Electricity | MSE ↓ | $\mathbf{0.172}$ | $0.193$ |
| | MAE ↓ | $\mathbf{0.321}$ | $0.345$ |
| RNNs on Pems-bay | MSE ↓ | $\mathbf{0.178}$ | $0.198$ |
| | MAE ↓ | $\mathbf{0.224}$ | $0.251$ |

We report the performance comparison between DLL and DLL-FA in Table 3. The results indicate that DLL generally outperforms DLL-FA across a wide range of tasks. On simple datasets like MNIST, DLL achieves a marginally higher accuracy (97.57%) compared to DLL-FA (97.37%), suggesting that the benefits of DLL are modest in such cases. However, on more complex datasets like CIFAR-10, DLL consistently demonstrates superior performance, with accuracy improvements of up to one percentage point.

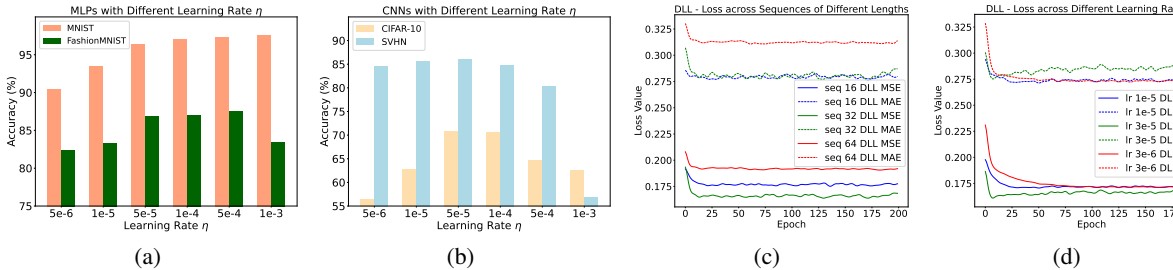

*Figure 3.* (a) MLPs trained with DLL by various learning rates. (b) CNNs trained with DLL by various learning rates. (c) Loss curves of RNNs trained with DLL across different sequence lengths. (d) Loss curves of RNNs trained with DLL by different learning rates.

Notably, in sequence modeling tasks, such as RNNs-based predictions on Harry Potter, DLL outperforms DLL-FA by a significant margin, with DLL-FA failing to converge in both cases. Furthermore, in time-series forecasting tasks, DLL demonstrates consistent superiority over DLL-FA, achieving lower MSE and MAE across the majority of datasets. These results highlight the critical role of updating $\Theta$ in DLL, which is essential for achieving better performance in both classification and sequential modeling tasks.

### 4.5. Analysis

In this section, we perform a convergence and sensitivity analysis for models trained with DLL. Since we update both the weights $\mathbf{W}$ and the special parameters $\Theta$ simultaneously during training with DLL, the learning rate $\eta$ plays a critical role in maintaining a balance between the trainable parameters during updates. Figures 3 (a) and (b) present the performance of MLPs and CNNs trained with DLL across various learning rates, respectively. We observe that the best performance for most models occurs around $\eta = 1 \times 10^{-4}$, indicating an optimal learning rate for stable training. Figures 3 (c) and (d) were evaluated using the Metr-la dataset. In Figure 3 (c), we plot the loss curves of RNNs trained with DLL across different sequence lengths. This suggests that excessively long or short sequence lengths may hinder convergence, making the training process unstable. Finally, Figure 3 (d) demonstrates how RNNs trained with DLL are affected by the learning rate. We find that a learning rate of $1 \times 10^{-4}$ is too large, leading to suboptimal performance, likely due to instability in gradient updates. This analysis highlights the importance of carefully tuning the learning rate for stable and effective training across different network architectures and tasks. In Appendix G, we show the time consumption and memory usage of the DLL in image classification tasks. In Appendix H, we evaluate the scalability of our method.

## 5. Related Work

Backpropagation (Rumelhart et al., 1986) has inherent limitations in terms of biological plausibility. It requires sym-

metric weight updates across layers, global error signals, and two-stage training, which are not naturally present in biological neural circuits. Currently, many attempts have been made to bridge this gap by exploring biologically plausible learning algorithms that mimic brain mechanisms, as discussed in Section 2.2. While there have been reviews (Weed & Hursting, 1998; Jiao et al., 2022; Millidge et al., 2022; Li et al., 2024; Schmidgall et al., 2024) summarizing the strengths, weaknesses, and differences between past learning algorithms, few have empirically benchmarked these algorithms on real-world datasets. In contrast, our study empirically evaluates these biologically plausible algorithms across various network architectures and datasets, providing a more comprehensive comparison.

While apical dendrites have been discussed in previous literature for credit assignment and have been utilized for various purposes, our study takes advantage of their properties to design and implement more biologically plausible learning algorithms, which differ significantly from existing approaches. Guerguiev et al. (2017) primarily aimed to explain how deep learning can be achieved using segregated dendritic compartments, but they did not propose a specific learning algorithm. As for Bartunov et al. (2018), while their proposed STDP improves upon DTP in performance and biological plausibility, it fails to resolve the issue of DTP requiring upper layers to propagate two separate values at different times. Payeur et al. (2021) investigated burst-dependent synaptic plasticity. While their work shares conceptual similarities with ours in terms of apical dendritic processing, the primary objective of their study differs from ours. Our proposed method not only satisfies all three criteria but also achieves higher performance compared to existing biologically plausible learning methods.

In addition, we aim for our proposed DLL to possess general capabilities comparable to those of BP, including the ability to perform both classification and regression tasks, handle diverse modalities such as images and language, and support multi-layer credit assignment. For example, in addition to image recognition tasks, our DLL framework can also be applied to train RNNs for regression tasks (Appendix B), such

as next-character prediction and time-series forecasting. In contrast, methods like BurstProp (Payeur et al., 2021) and SoftHebb (Journé et al., 2023) may struggle with such tasks, as their designs are not well-suited for recurrent architectures. Counter-current Learning (Kao & Hariharan, 2024) violates our third criterion, as it explicitly involves distinct forward and backward phases.

## 6. Conclusion

In this paper, we reviewed the current biologically plausible learning algorithms and summarized three criteria that an ideal learning algorithm should satisfy. Meanwhile, we empirically evaluate these algorithms across diverse network architectures and datasets. Secondly, we introduced Dendritic Localized Learning (DLL), a novel learning algorithm designed to meet these criteria while maintaining the effectiveness of training MLPs, CNNs, and RNNs. Finally, to validate its performance, we present extensive experimental results across a range of tasks, including image recognition, text character prediction, and time-series forecasting, utilizing MLPs, CNNs, and RNNs. By combining theoretical rigor with practical applicability, our work paves the way for future research into biologically plausible learning paradigms, fostering deeper connections between neuroscience and artificial intelligence. The limitations and future directions are discussed in Appendix I.

## Impact Statement

This work bridges the gap between neuroscience and artificial intelligence by introducing a biologically plausible learning algorithm that achieves competitive performance with backpropagation across diverse tasks. By fostering advancements in brain-inspired computing, our research opens new pathways for developing bio-interpretable and scalable machine learning models. We do not think our work will negatively impact ethical aspects or future societal consequences.

## Acknowledgments

The authors would like to thank the anonymous reviewers for their valuable comments. This work was supported by the National Natural Science Foundation of China (No. 62076068).

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

## A. Global Algorithm of Dendritic Localized Learning

In this section, we show the global algorithm of dendritic localized learning in a pseudo-code style.

---

**Algorithm 1** Algorithm of Dendritic Localized Learning

---

**Input:** data $D$, number of layers $L$, number of neurons per layer $N$, learning rate $\eta_{\mathbf{W}}, \eta_{\mathbf{\Theta}}$

Initialize $\mathbf{W}, \mathbf{\Theta}$ randomly for all layers

**for** epoch = 0 to max_epochs **do**

  **for** each batch $\mathbf{B}$ in $D$ **do**

    **Forward Pass** (when input stimuli are fed to the basal dendrite):

    Assign the input values to the basal dendrites of the input layer neurons: $\mathbf{u}_0 = \mathbf{B}$, and initialize $\mathbf{x}_0 = \mathbf{u}_0$

    **for** $i = 0$ to $L - 1$ **do**

      Compute forward pass $\mathbf{u}_{i+1} = f_i(\mathbf{W}_i\mathbf{u}_i)$, where $f_i$ is the activation function for the $i$-th layer

      Initialize $\mathbf{x}_{i+1} = \mathbf{u}_{i+1}$

    **end for**

    **Compute Local Errors** (when apical dentrite receives the top-down feedback):

    Assign the target values to the apical dendrites of the output layer neurons: $\mathbf{x}_L = \mathbf{target}$

    Compute output error $\xi_L = -\nabla_{\mathbf{u}_L}\mathcal{L}(\mathbf{u}_L, \mathbf{x}_L)$

    Compute input error $\xi_0 = \mathbf{x}_0 - \mathbf{u}_0$

    **for** $i = L - 1$ down to 1 **do**

      Compute local error $\xi_i = \mathbf{\Theta}_i^T[\xi_{i+1} \odot f_i{}'(\mathbf{W}_i\mathbf{u}_i)]$

    **end for**

    **Update Weights and Thetas Simultaneously:**

    **for** $i = 0$ to $L - 1$ **do**

      **if** $\boldsymbol{\xi}_{i+1} \neq 0$ **then**

        Update $\mathbf{W}_i \leftarrow \mathbf{W}_i + \eta_{\mathbf{W}} \cdot (\boldsymbol{\xi}_{i+1} \odot f_i{}'(\mathbf{W}_i\mathbf{u}_i)\mathbf{u}_i)$

        Update $\mathbf{\Theta}_i \leftarrow \mathbf{\Theta}_i + \eta_{\mathbf{\Theta}} \cdot \left\{-\boldsymbol{\xi}_i\left[\boldsymbol{\xi}_{i+1} \odot f_i{}'(\mathbf{W}_i\mathbf{u}_i)\right]\right\}^T$

      **end if**

    **end for**

  **end for**

**end for**

---

## B. Training Recurrent Neural Networks with Dendritic Localized Learning

In the conventional Recurrent Neural Networks (RNNs) framework, the hidden layer consists of a single vector, denoted as $\mathbf{h}$. However, our DLL-RNNs model is inspired by the structure of pyramidal neurons, incorporating two distinct components within the hidden layer. One component, $\mathbf{h}^s$, represents sensory input from the basal of the pyramidal neuron, while the other, $\mathbf{h}^p$, denotes the backpropagated activity from the apical of the pyramidal neuron. Similar to the standard RNNs framework, the DLL-RNNs operates across multiple time steps. At the $i$-th time step, the hidden state in RNNs is represented as $\mathbf{h}_i$, whereas in the DLL-RNNs, it is split into two components: $\mathbf{h}_i^s$ and $\mathbf{h}_i^p$. Specifically, at each time step, we first compute the value of $\mathbf{h}_i^s$, which is then used to initialize $\mathbf{h}_i^p$ as:

$$\mathbf{h}_i^p = \mathbf{h}_i^s = f(\mathbf{W_h}\mathbf{h}_{i-1}^s + \mathbf{W_x}\mathbf{x}_i). \tag{10}$$

Here, $\mathbf{x}_i$ is the input at time step $i$ for both the traditional RNNs and the DLL-RNNs. The computation involves multiplying the weight matrix $\mathbf{W_h}$ by the sensory component of the hidden state from the previous time step, $\mathbf{h}_{i-1}^s$, and adding it to the product of the weight matrix $\mathbf{W_x}$ and the input $\mathbf{x}_i$ for the current time step in our DLL-RNNs. This sum is then processed through the activation function $f$, resulting in the hidden state for the current time step, $\mathbf{h}_i^s$. We define the local error $\boldsymbol{\xi}_i^{\mathbf{h}}$ of the hidden layer in the $i$-th time step as:

$$\boldsymbol{\xi}_i^{\mathbf{h}} = \mathbf{h}_i^p - \mathbf{h}_i^s. \tag{11}$$

We use $\mathbf{t}_i$ to represent the expected output for the $i$-th time step, and use $\mathbf{y}_i$ to denote the actual output for the $i$-th time step.

At time step $i$, the output of the DLL-RNNs is given by $\mathbf{y}_i$, which is obtained by multiplying the weight matrix $\mathbf{W_y}$ with the hidden state $\mathbf{h}_i^s$, followed by the application of an activation function $g$ (In our DLL-RNNs, the function $g$ is a linear function defined as $g(x) = x$), written as:

$$\mathbf{y}_i = g(\mathbf{W_y}\mathbf{h}_i^s) = \mathbf{W_y}\mathbf{h}_i^s. \tag{12}$$

And We define $\boldsymbol{\xi}_i^{\mathbf{y}}$ as the error for the output layer in the $i$-th time step.

$$\boldsymbol{\xi}_i^{\mathbf{y}} = \mathbf{t}_i - \mathbf{y}_i. \tag{13}$$

In our DLL-RNNs, we construct the Mean Squared Error (MSE) loss using the hidden layer and the output layer at each time step $i$, denoted as $\boldsymbol{\xi}_i^{\mathbf{h}}$ and $\boldsymbol{\xi}_i^{\mathbf{y}}$. The MSE loss is expressed as:

$$\mathcal{L} = -\frac{1}{2}\sum_{i=1}^{n}[(\boldsymbol{\xi}_i^{\mathbf{h}})^2 + (\boldsymbol{\xi}_i^{\mathbf{y}})^2] = -\frac{1}{2}\sum_{i=1}^{n}\left[(\mathbf{h}_i^p - \mathbf{h}_i^s)^2 + (\mathbf{t}_i - \mathbf{y}_i)^2\right]. \tag{14}$$

To compute the backpropagated activity $\mathbf{h}_i^p$, we differentiate the loss $\mathcal{L}$ with respect to $\mathbf{h}_i^p$ to derive $\Delta\mathbf{h}_i^p$, which is the direction of change for $\mathbf{h}_i^p$. We update $\mathbf{h}_i^p$ through $\mathbf{h}_i^p \leftarrow \mathbf{h}_i^p + \eta_{\mathbf{h}} \cdot \Delta\mathbf{h}_i^p$, where $\eta_{\mathbf{h}}$ represents the learning rate for updating $\mathbf{h}_i^p$. For the time step $i \in [1, n-1]$, $\Delta\mathbf{h}_i^p$ is defined as:

$$
\begin{aligned}
\Delta\mathbf{h}_i^p &= \frac{\partial\mathcal{L}}{\partial\mathbf{h}_i^p} \\
&= \frac{\partial\{-\frac{1}{2}\sum_{j=i}^{n}\left[(\mathbf{h}_j^p - \mathbf{h}_j^s)^2 + (\mathbf{t}_j - \mathbf{y}_j)^2\right]\}}{\partial\mathbf{h}_i^p} \\
&= \frac{\partial\left[-\frac{1}{2}(\mathbf{h}_i^p - \mathbf{h}_i^s)^2 - \frac{1}{2}(\mathbf{t}_i - \mathbf{y}_i)^2\right]}{\partial\mathbf{h}_i^p} + \frac{\partial\{-\frac{1}{2}\sum_{j=i+1}^{n}\left[(\mathbf{h}_j^p - \mathbf{h}_j^s)^2 + (\mathbf{t}_j - \mathbf{y}_j)^2\right]\}}{\partial\mathbf{h}_{i+1}^p}\frac{\partial\mathbf{h}_{i+1}^p}{\partial\mathbf{h}_i^p} \\
&= -\boldsymbol{\xi}_i^{\mathbf{h}} + \frac{\partial\mathcal{L}}{\partial\mathbf{y}_i}\frac{\partial\mathbf{y}_i}{\partial\mathbf{h}_i^p} + \frac{\partial\mathcal{L}}{\partial\mathbf{h}_{i+1}^p}\frac{\partial\mathbf{h}_{i+1}^p}{\partial\mathbf{h}_i^p} \\
&= -\boldsymbol{\xi}_i^{\mathbf{h}} + \frac{\partial\left[-\frac{1}{2}(\mathbf{t}_i - \mathbf{y}_i)^2\right]}{\partial\mathbf{y}_i}\frac{\partial g(\mathbf{W_y}\mathbf{h}_i^p)}{\partial\mathbf{h}_i^p} + \Delta\mathbf{h}_{i+1}^p\frac{\partial\mathbf{h}_{i+1}^p}{\partial\mathbf{h}_i^p} \\
&= -\boldsymbol{\xi}_i^{\mathbf{h}} + \mathbf{W_y}^T\left[\boldsymbol{\xi}_i^{\mathbf{y}} \odot g'(\mathbf{W_y}\mathbf{h}_n^s)\right] + \mathbf{W_h}^T\left[\boldsymbol{\xi}_{i+1}^{\mathbf{h}} \odot f'(\mathbf{W_h}\mathbf{h}_i^p + \mathbf{W_x}\mathbf{x}_{i+1})\right] \\
&= -\boldsymbol{\xi}_i^{\mathbf{h}} + \mathbf{W_y}^T\boldsymbol{\xi}_i^{\mathbf{y}} + \mathbf{W_h}^T\left[\boldsymbol{\xi}_{i+1}^{\mathbf{h}} \odot f'(\mathbf{W_h}\mathbf{h}_i^p + \mathbf{W_x}\mathbf{x}_{i+1})\right].
\end{aligned} \tag{15}
$$

For time step $i = n$, the formula for $\Delta\mathbf{h}_n^p$ is given as:

$$
\begin{aligned}
\Delta\mathbf{h}_n^p &= \frac{\partial\mathcal{L}}{\partial\mathbf{h}_n^p} \\
&= \frac{\partial\left[-\frac{1}{2}(\mathbf{h}_n^p - \mathbf{h}_n^s)^2 - \frac{1}{2}(\mathbf{t}_n - \mathbf{y}_n)^2\right]}{\partial\mathbf{h}_n^p} \\
&= -(\mathbf{h}_n^p - \mathbf{h}_n^s) - (\mathbf{t}_n - \mathbf{y}_n)(-\frac{\partial\mathbf{y}_n}{\partial\mathbf{h}_n^p}) \\
&= -\boldsymbol{\xi}_n^{\mathbf{h}} + \mathbf{W_y}^T\left[\boldsymbol{\xi}_n^{\mathbf{y}} \odot g'(\mathbf{W_y}\mathbf{h}_n^s)\right] \\
&= -\boldsymbol{\xi}_n^{\mathbf{h}} + \mathbf{W_y}^T\boldsymbol{\xi}_n^{\mathbf{y}}.
\end{aligned} \tag{16}
$$

Here, $\odot$ denotes the Hadamard product, and since $g$ is a linear function defined as $g(x) = x$, its derivative is $g'(x) = 1$. Thus, $g'(\mathbf{W_y}\mathbf{h}_n^s) = 1$. In our DLL-RNNs, similar to the DLL, we replace $\mathbf{W_y}$ with $\boldsymbol{\Theta}_{\mathbf{y}}$ and $\mathbf{W_h}$ with $\boldsymbol{\Theta}_{\mathbf{h}}$. However, $\mathbf{W_x}$ does not appear in the computation of $\boldsymbol{\xi}_i^{\mathbf{h}}$, so there is no need for replacement in this context.

In our DLL-RNNs, we updates $\Delta\mathbf{h}_i^p$ as:

$$\Delta\mathbf{h}_i^p = -\boldsymbol{\xi}_i^{\mathbf{h}} + \boldsymbol{\Theta}_{\mathbf{y}}^T\boldsymbol{\xi}_i^{\mathbf{y}} + \boldsymbol{\Theta}_{\mathbf{h}}^T\left[\boldsymbol{\xi}_{i+1}^{\mathbf{h}} \odot f'(\mathbf{W_h}\mathbf{h}_i^p + \mathbf{W_x}\mathbf{x}_{i+1})\right]. \tag{17}$$

As well, we update $\Delta\mathbf{h}_n^p$ as:

$$\Delta\mathbf{h}_n^p = -\boldsymbol{\xi}_n^{\mathbf{h}} + \boldsymbol{\Theta}_{\mathbf{y}}^T\boldsymbol{\xi}_n^{\mathbf{y}}. \tag{18}$$

Similar to DLL-MLPs, we can directly assign the value of $\boldsymbol{\xi}_i^{\mathbf{h}}$ as:

$$\boldsymbol{\xi}_i^{\mathbf{h}} = \boldsymbol{\Theta}_{\mathbf{y}}^T \boldsymbol{\xi}_i^{\mathbf{y}} + \boldsymbol{\Theta}_{\mathbf{h}}^T [\boldsymbol{\xi}_{i+1}^{\mathbf{h}} \odot f'(\mathbf{W_h} \mathbf{h}_i^s + \mathbf{W_x} \mathbf{x}_{i+1})]. \tag{19}$$

And assign $\boldsymbol{\xi}_n^{\mathbf{h}}$ as:

$$\boldsymbol{\xi}_n^{\mathbf{h}} = \boldsymbol{\Theta}_{\mathbf{y}}^T \boldsymbol{\xi}_n^{\mathbf{y}}. \tag{20}$$

Finally, we will adjust $\mathbf{W}$ through $\mathbf{W} \leftarrow \mathbf{W} + \eta * \Delta \mathbf{W}$. The symbol $\eta$ denotes the learning rate for various instances of the weight matrix $\mathbf{W}$. Detailed update rules for the weight matrices $\Delta \mathbf{W_y}$ are defined as:

$$\begin{aligned}
\Delta \mathbf{W_y} &= \frac{\partial \mathcal{L}}{\partial \mathbf{W_y}} \\
&= \sum_{i=1}^n \frac{\partial \mathcal{L}}{\partial \mathbf{y}_i} \frac{\partial \mathbf{y}_i}{\partial \mathbf{W_y}} \\
&= \sum_{i=1}^n (\mathbf{t}_i - \mathbf{y}_i)(\mathbf{h}_i^s)^T \\
&= \sum_{i=1}^n \boldsymbol{\xi}_i^{\mathbf{y}}(\mathbf{h}_i^s)^T.
\end{aligned} \tag{21}$$

Then the $\mathbf{W_y}$ will be updated as:

$$\mathbf{W_y} \leftarrow \mathbf{W_y} + \eta_{\mathbf{W_y}} * \Delta \mathbf{W_y}. \tag{22}$$

For $\Delta \mathbf{W_x}$, we calculate as:

$$\begin{aligned}
\Delta \mathbf{W_x} &= \frac{\partial \mathcal{L}}{\partial \mathbf{W_x}} = \sum_{i=1}^n \frac{\partial \mathcal{L}}{\partial \mathbf{h}_i^s} \frac{\partial \mathbf{h}_i^s}{\partial \mathbf{W_x}} \\
&= \sum_{i=1}^n [\frac{\partial \mathcal{L}}{\partial \mathbf{h}_i^s} \odot f'(\mathbf{W_h} \mathbf{h}_{i-1}^s + \mathbf{W_x} \mathbf{x}_i)] \mathbf{x}_i \\
&= \sum_{i=1}^n [\boldsymbol{\xi}_i^{\mathbf{h}} \odot f'(\mathbf{W_h} \mathbf{h}_{i-1}^s + \mathbf{W_x} \mathbf{x}_i)] \mathbf{x}_i.
\end{aligned} \tag{23}$$

Then the $\Delta \mathbf{W_x}$ will be updated as:

$$\mathbf{W_x} \leftarrow \mathbf{W_x} + \eta_{\mathbf{W_x}} * \Delta \mathbf{W_x}. \tag{24}$$

And for $\Delta \mathbf{W_h}$, we update it as:

$$\begin{aligned}
\Delta \mathbf{W_h} &= \frac{\partial \mathcal{L}}{\partial \mathbf{W_h}} \\
&= \sum_{i=1}^n \frac{\partial \mathcal{L}}{\partial \mathbf{h}_i^s} \frac{\partial \mathbf{h}_i^s}{\partial \mathbf{W_h}} \\
&= \sum_{i=1}^n [\boldsymbol{\xi}_i^{\mathbf{h}} \odot f'(\mathbf{W_h} \mathbf{h}_{i-1}^s + \mathbf{W_x} \mathbf{x}_i)] \mathbf{h}_{i-1}^s.
\end{aligned} \tag{25}$$

We update $\mathbf{W_h}$ as:

$$\mathbf{W_h} \leftarrow \mathbf{W_h} + \eta_{\mathbf{W_h}} * \Delta \mathbf{W_h}. \tag{26}$$

Similarly, the parameter vector $\boldsymbol{\Theta}$ is updated using the following equation: $\boldsymbol{\Theta} \leftarrow \boldsymbol{\Theta} + \eta * \Delta \boldsymbol{\Theta}$. Here, $\eta$ represents the learning rate, which varies for different instances of the parameter vector $\boldsymbol{\Theta}$. The update term $\Delta \boldsymbol{\Theta}$ and the specific update

formula are defined as:

$$
\begin{aligned}
\Delta\boldsymbol{\Theta}_{\mathbf{y}}^{T} &= \frac{\partial\mathcal{L}}{\partial\boldsymbol{\Theta}_{\mathbf{y}}^{T}} \\
&= \sum_{i=1}^{n}\frac{\partial\mathcal{L}}{\partial\boldsymbol{\xi}_{i}^{\mathbf{h}}}\frac{\partial\boldsymbol{\xi}_{i}^{\mathbf{h}}}{\partial\boldsymbol{\Theta}_{\mathbf{y}}^{T}} \\
&= \sum_{i=1}^{n}-\boldsymbol{\xi}_{i}^{\mathbf{h}}[\boldsymbol{\xi}_{i}^{\mathbf{y}}\odot g'(\mathbf{W}_{\mathbf{y}}\mathbf{h}_{n}^{s})]^{T} \\
&= \sum_{i=1}^{n}-\boldsymbol{\xi}_{i}^{\mathbf{h}}(\boldsymbol{\xi}_{i}^{\mathbf{y}})^{T}.
\end{aligned}
\tag{27}
$$

Then $\boldsymbol{\Theta}_{\mathbf{y}}^{T}$ will be updated as:

$$
\boldsymbol{\Theta}_{\mathbf{y}} \leftarrow \boldsymbol{\Theta}_{\mathbf{y}} + \eta_{\boldsymbol{\Theta}_{\mathbf{y}}} * \Delta(\boldsymbol{\Theta}_{\mathbf{y}}^{T})^{T}.
\tag{28}
$$

For $\Delta\boldsymbol{\Theta}_{\mathbf{h}}^{T}$, we calculate it as:

$$
\Delta\boldsymbol{\Theta}_{\mathbf{h}}^{T} = \sum_{i=1}^{n-1}-\boldsymbol{\xi}_{i}^{\mathbf{h}}[\boldsymbol{\xi}_{i+1}^{\mathbf{h}}\odot f'(\mathbf{W}_{\mathbf{h}}\mathbf{h}_{i}^{s}+\mathbf{W}_{\mathbf{x}}\mathbf{x}_{i+1})]^{T}.
\tag{29}
$$

So the $\boldsymbol{\Theta}_{\mathbf{h}}$ will be:

$$
\boldsymbol{\Theta}_{\mathbf{h}} \leftarrow \boldsymbol{\Theta}_{\mathbf{h}} + \eta_{\boldsymbol{\Theta}_{\mathbf{h}}} * \Delta(\boldsymbol{\Theta}_{\mathbf{h}}^{T})^{T}.
\tag{30}
$$

## C. Discussion on Convergence

The loss function (Equation (3)) is designed to minimize the discrepancy between the top-down predictions and bottom-up outputs of each pyramidal neuron in the network. To achieve this, we employ local gradient descent–based learning rules and neural plasticity mechanisms to update both forward and backward weights. During each iteration, the differences between the network's predictions and the ground truth propagate back through localized errors, effectively coordinating all neurons in an orchestrated manner. As a result, neural responses collectively refine predictions over successive iterations, gradually reducing local errors and driving the network toward convergence. While providing formal convergence proofs remains challenging due to the network's nonlinear operations, our empirical results consistently demonstrate a steady decrease in loss throughout training, supporting the stability and effectiveness of our approach.

## D. Experiment Settings

### D.1. Statistics of Datasets

For the image classification task, we utilize several widely used image recognition datasets to evaluate our model's performance across different domains. These datasets include MNIST, FashionMNIST, SVHN, and CIFAR-10.

- **MNIST.** The MNIST dataset consists of 70,000 grayscale images of handwritten digits, each with a resolution of 28x28 pixels. It is a classic benchmark for evaluating the performance of machine learning models on digit classification tasks, with 60,000 training images and 10,000 test images.

- **FashionMNIST.** FashionMNIST is a dataset similar to MNIST but consists of images of clothing items, such as shirts, pants, and shoes. It contains 60,000 training images and 10,000 test images, each with a resolution of 28x28 pixels. FashionMNIST serves as a more complex alternative to MNIST for testing models on multi-class image classification tasks.

- **Street View House Numbers (SVHN).** SVHN is a dataset that consists of over 600,000 labeled digits extracted from street-level images of house numbers. The dataset includes images of size 32x32 pixels in three color channels (RGB). SVHN is designed for digit recognition in real-world, natural scene contexts, making it more challenging than MNIST.

- **CIFAR-10.** CIFAR-10 is a dataset comprising 60,000 32x32 color images in 10 different categories, with 6,000 images per category. The dataset is split into 50,000 training images and 10,000 test images, representing a range of objects including airplanes, automobiles, and animals. CIFAR-10 is widely used for benchmarking models in object recognition tasks.

For the text character prediction task, we utilize the Harry Potter Series to evaluate our model's performance..

- **Harry Potter Series.** This dataset includes the entire text of the Harry Potter book series, providing a unique context for text character prediction tasks. It is particularly useful for analyzing themes, character relationships, and genre-specific language, allowing models to be evaluated on their ability to understand narrative structures and stylistic elements.

For the time-series forecasting task, we utilize the Electricity, Metr-la, and Pems-bay.

- **Electricity.** The dataset contains hourly electricity consumption data from 321 clients, covering three years from 2012 to 2014. It records 15-minute interval values in kilowatts, with no missing data. Each column represents a client, and consumption is set to zero before a client's start date.

- **Metro Traffic Los Angeles (Metr-la).** The Metr-la dataset consists of traffic speed data collected from 207 loop detectors on the highways of Los Angeles. It provides measurements recorded every 5 minutes, capturing temporal patterns in traffic dynamics.

- **Pems-bay.** The Pems-bay dataset is sourced from the Performance Measurement System (PeMS) maintained by California Transportation Agencies (CalTrans). It includes traffic data collected from 325 sensors located in the Bay Area. The dataset spans six months, from January 1, 2017, to May 31, 2017, providing a detailed temporal view of traffic conditions.

## D.2. Implementation Details

### D.2.1. DLL-MLPs

**Model Architecture**     Given the distinct input dimensions and varying levels of classification complexity across the four experimental datasets, we designed tailored architectures for each dataset. Specifically, the MNIST and FashionMNIST datasets consist of single-channel images with a size of 28×28 pixels, whereas the SVHN and CIFAR-10 datasets comprise three-channel color images of 32×32 pixels. To address these differences, we adapted the model architectures to optimize performance. For the MNIST and FashionMNIST tasks, we employed 5-layer DLL-MLPs with layer sizes of 784, 1024, 512, 256, and 10 neurons, respectively. The initial layer size of 784 corresponds to the flattened input vector from the 28×28 single-channel images, while the final layer of 10 neurons represents the output classes. In contrast, for the more complex SVHN and CIFAR-10 datasets, we utilized a deeper 6-layer DLL-MLPs, with layer sizes of 3072, 4096, 2048, 1024, 256, and 10 neurons. The first layer size of 3072 reflects the flattened input vector from the 32×32×3 three-channel images, and the last layer also comprises 10 neurons for the classification outputs.

**Training Hyper-parameters**     All models were optimized using the Adam optimizer, with a linear learning rate scheduler for weight decay. The hyperbolic tangent activation function was employed in all layers except the output layer. Given the varying sizes and complexities of the datasets, we tailored the hyperparameter configurations to achieve optimal performance for each dataset. For the MNIST dataset, we set the learning rate to $1 \times 10^{-3}$ and used a batch size of 128. For FashionMNIST, considering its increased complexity relative to MNIST, we adjusted the learning rate to $5 \times 10^{-4}$ and used a batch size of 64 to better handle the more nuanced classification task. For the SVHN dataset, which presents a higher level of complexity, we set the learning rate to $5 \times 10^{-5}$ and maintained a batch size of 64 to balance training stability and convergence speed. Finally, for CIFAR-10, the most challenging dataset among the four, we set the learning rate to $8 \times 10^{-5}$ and also used a batch size of 64 to ensure robust training dynamics.

### D.2.2. DLL-CNNs

**Model Architecture**     Following the same rationale as in the DLL-MLPs model, we employed different architectures for different datasets.

For the MNIST dataset, the first layer is a convolutional layer with 32 filters, a kernel size of 5, and no padding. The second layer is a max-pooling layer. The third layer is a convolutional layer with 64 filters, a kernel size of 3, and no padding. The fourth layer is another max-pooling layer. The fifth layer is a convolutional layer with 16 filters, a kernel size of 3, and no padding. The sixth layer is a projection layer with an output size of 200. The seventh and final layer is the output layer, with an output size of 10.

For the FashionMNIST dataset, the first layer is a convolutional layer with 64 filters, a kernel size of 5, and no padding. The second layer is a max-pooling layer. The third layer is a convolutional layer with 128 filters, a kernel size of 3, and no padding. The fourth layer is an average pooling layer. The fifth layer is a convolutional layer with 64 filters, a kernel size of 3, and no padding. The sixth layer is a projection layer with an output size of 128. The seventh and final layer is the output layer, with an output size of 10.

For the SVHN dataset, the first layer is a convolutional layer with 64 filters, a kernel size of 3, and padding of 1. The second layer is a max-pooling layer. The third layer is a convolutional layer with 128 filters, a kernel size of 3, and padding of 1. The fourth layer is another max-pooling layer. The fifth layer is a convolutional layer with 64 filters, a kernel size of 3, and padding of 1. The sixth layer is a projection layer with an output size of 256. The seventh and final layer is the output layer, with an output size of 10.

For the CIFAR-10 dataset, the first layer is a convolutional layer with 64 filters, a kernel size of 3, and padding of 1. The second layer is a max-pooling layer. The third layer is a convolutional layer with 128 filters, a kernel size of 3, and padding of 1. The fourth layer is another max-pooling layer. The fifth layer is a convolutional layer with 64 filters, a kernel size of 3, and padding of 1. The sixth layer is an average pooling layer. The seventh layer is a projection layer with an output size of 256. The eighth and final layer is the output layer, with an output size of 10.

**Training Hyper-parameters**    All models were optimized using the Adam optimizer, with a linear learning rate scheduler for weight decay. The hyperbolic tangent activation function was employed in all layers except the output layer. For consistency across all datasets, we standardized the hyperparameters to a learning rate of $5 \times 10^{-5}$ and a batch size of 64.

### D.2.3. DLL-RNNs

**Model Architecture**    Recurrent Neural Networks (RNNs) are a class of neural networks designed to recognize patterns in sequences of data, such as time series, natural language, or any other sequence data. Unlike traditional feedforward neural networks, RNNs have connections that form directed cycles, allowing them to maintain a hidden state that can capture information about previous time steps. In our DLL-RNNs architecture, the network consists of three layers:

- **Input Layer**: This layer accepts a one-dimensional tensor. The dimensionality of this tensor is determined by the data format of different datasets.

- **Hidden Layer**: The hidden layer's dimensionality is a tunable hyperparameter. We have implemented a DLL version of the RNNs, where the hidden layer is divided into two parts: $\mathbf{h}_i^s$, responsible for forward output, and $\mathbf{h}_i^p$, responsible for receiving error signals.

- **Output Layer**: This layer outputs a one-dimensional tensor, with its dimensionality also determined by the specific dataset's requirements.

In our DLL-RNNs model, during the output phase at time step $i$, the $\mathbf{h}_i^s$ part of the hidden layer receives input $x$ and the input from the previous time step $\mathbf{h}_{i-1}^s$. It then outputs to the current time step $\mathbf{y}_i$ and the next time step $\mathbf{h}_{i+1}^s$.

**Evaluation Metrics**    We utilize character prediction accuracy to evaluate the performance of the DLL-RNNs model on a text character prediction task. Given a character sequence of length $n + 1$, denoted as $[0, n]$, the model is designed to operate over $n$ time steps. During each time step $i$, the model receives a one-hot encoded tensor corresponding to the $i$-th character and predicts the one-hot encoded tensor for the $(i + 1)$-th character. This process involves iteratively accepting input and generating predictions. The evaluation metric is based on counting the number of correct predictions made at each step, thereby measuring how well the model learns the sequential patterns and predicts subsequent characters accurately throughout the sequence. We utilize the Mean Squared Error (MSE) loss and the Mean Absolute Error (MAE) loss to evaluate the performance of the DLL-RNNs model in the context of time-series forecasting tasks. The Mean Squared Error

(MSE) is defined as:

$$\text{MSE} = \frac{1}{n} \sum_{i=1}^{n} (y_i - \hat{\mathbf{y}}_i)^2. \tag{31}$$

In this formula, $n$ represents the total number of observations. The symbol $y_i$ denotes the true value of the $i$-th observation, while $\hat{\mathbf{y}}_i$ stands for the backpropagated label of the $i$-th observation. The term $(y_i - \hat{\mathbf{y}}_i)^2$ is the squared difference between the true and backpropagated labels for the $i$-th observation. The Mean Absolute Error (MAE) is defined as:

$$\text{MAE} = \frac{1}{n} \sum_{i=1}^{n} |y_i - \hat{\mathbf{y}}_i|. \tag{32}$$

Here, $n$ is the total number of observations. The variable $y_i$ indicates the true value of the $i$-th observation, and $\hat{\mathbf{y}}_i$ represents the predicted value of the $i$-th observation. The expression $|y_i - \hat{\mathbf{y}}_i|$ refers to the absolute difference between the true and backpropagated labels for the $i$-th observation.

**Training Hyper-parameters**    We conducted a grid search on each dataset to explore the optimal parameter configurations. Specifically, we selected appropriate hidden layer sizes and learning rates tailored to different tasks to achieve the best training outcomes. Additionally, we employed both cosine and linear schedulers to dynamically adjust the learning rate based on the epoch. This approach ensures that the learning process adapts effectively throughout the training phases, optimizing the model's performance across various datasets and tasks. Ultimately, we configured the hidden layer sizes for different datasets as follows: for the Harry Potter Series, the hidden layer size was set to 324; for the Electricity and Metr-la dataset, it was set to 300; and for the Pems-bay dataset, the hidden layer size was set to 384.

## E. Experimental Results on CIFAR-100 and Tiny-ImageNet

We trained CNNs with BP and DLL on CIFAR-100 and TinyImageNet. Consistent with previous work (Bartunov et al., 2018), we report test accuracy for CIFAR-100 and test error rate for Tiny-ImageNet in Table 4.

*Table 4.* Performance of CNNs trained with BP and DLL on CIFAR-100 and Tiny-ImageNet. Test accuracy for CIFAR-100 and test error rate for Tiny-ImageNet.

| Method | CIFAR-100 | Tiny-ImageNet |
|---|---|---|
| BP_CNN | **44.50**% | **78.60**% |
| DLL_CNN | 38.60% | 82.90% |

Note that we do not use any additional training techniques such as batch normalization or residual connections. Our CNN architecture is similar to that in Bartunov et al. (2018). For CIFAR-100, the CNN consists of four convolutional layers with channel configurations of 3-64-64-128-64, followed by two fully connected layers. For TinyImageNet, the CNN consists of five convolutional layers with filter configurations of 3-64-64-128-128-64, followed by two fully connected layers.

## F. Training TextCNN with DLL

In this section, we performed experiments on the text classification datasets Subj[1] and Movie Review (MR) (Pang & Lee, 2005), with results summarized in Table 5.

*Table 5.* Performance of TextCNN on text classification tasks.

| Method | Subj | MR |
|---|---|---|
| BP_TextCNN | **88.50**% | **74.68**% |
| DLL_TextCNN | 84.40% | 70.79% |

The architecture is identical to the original TextCNN (Kim, 2014). The models trained with DLL achieve comparable accuracy to those trained with BP on two datasets.

---

[1] https://www.cs.cornell.edu/people/pabo/movie-review-data/

## G. Time Consumption and Memory Usage

We record the training time consumption and GPU memory usage when conducting experiments on image classification. The time consumption and memory usage for these experiments are summarized in Table 6:

*Table 6.* Training time consumption and GPU memory usage of BP and DLL.

| Method | Training Time (s/Epoch) | Memory Usage (MB) |
|---|---|---|
| BP_MLP | 31.6 | 1286.4 |
| BP_CNN | 99.0 | 1272.9 |
| DLL_MLP | 44.7 | 1595.3 |
| DLL_CNN | 169.8 | 1306.9 |

To fairly compare time consumption across architectures, we used the CPU instead of the GPU. DLL requires more training time and memory because both the forward weight $\mathbf{W}$ and backward weight $\Theta$ are updated simultaneously. Our design is not driven by computational or memory efficiency; rather, we prioritize biological plausibility.

## H. Scalability of DLL

In this section, we conduct scalability experiments and show the results in Table 7:

*Table 7.* Scalability of MLPs trained with DLL.

| DLL-MLP Architecture | Accuracy on MNIST |
|---|---|
| 784-1024-10 | 71.15% |
| 784-1024-512-10 | 89.61% |
| 784-1024-512-256-10 | 97.57% |

All MLPs are trained fairly, and the results show the scalability of DLL.

## I. Limitations and Future Directions

### I.1. Limitations.

Despite the promising contributions of Dendritic Localized Learning (DLL), several limitations remain. First, the implementation and evaluation of DLL have been restricted to conventional artificial neural networks (ANNs) architectures such as MLPs, CNNs, and RNNs, leaving its applicability to spiking neural networks (SNNs) (Maass, 1997) unexplored. Given the increasing interest in SNNs for energy-efficient and biologically realistic computing, this is a critical area for future work. Second, while DLL achieves biological plausibility by satisfying the proposed criteria, its reliance on unsigned error signals may present challenges in scenarios requiring precise error alignment or task-specific optimization. Addressing these issues is essential for improving both the versatility and robustness of the algorithm.

### I.2. Future directions.

Future research will focus on extending the DLL framework to SNNs, leveraging their potential for energy-efficient computation and neuromorphic hardware compatibility. This extension will require adapting the DLL to handle the temporal dynamics and discrete spike-based representations inherent in SNNs, further aligning the algorithm with biological principles. Additionally, exploring hybrid architectures that integrate DLL with traditional learning mechanisms may enhance their scalability and performance on more complex datasets and tasks. Another avenue involves addressing the challenges posed by unsigned error signals, such as developing task-specific adjustments or augmenting the algorithm with mechanisms to improve error precision. Finally, we envision applying DLL to real-world problems in areas like robotics, neuroscience, and autonomous systems, validating its practical utility and impact across diverse domains.

