# OpenReview forum: "Dendritic Localized Learning: Toward Biologically Plausible Algorithm"
_ICML.cc/2025/Conference — ICML 2025 poster_

### Official Review · Reviewer_Crph · 2025-03-08

**Overall Recommendation:** 2

**Summary:**

This work proposes a biologically plausible algorithm for training deep neural networks utilizing apical dendrites. The authors apply the proposed algorithms to learning in MLPs, CNNs, and RNNs, and demonstrate that the proposed algorithm outperforms previous biologically plausible algorithms that satisfy all three plausibility criteria set up by the authors.

**Claims And Evidence:**

The manuscript is technically sound except for the issues on evaluation and novelty discussed below.

**Essential References Not Discussed:**

Please see the comment above.

**Experimental Designs Or Analyses:**

Implementation details of previous algorithms are missing.

**Methods And Evaluation Criteria:**

The implementation of the previous algorithms appears to be suboptimal. For instance, in table 1, the performance of feedback alignment in MNIST with MLPs is 91.87%. However, previous work has shown that this algorithm can achieve 98% test performance (Bartunov et al., NeurIPS 2018).

Additionally, I found the second criterion for the biological plausibility to be somewhat problematic considering the ubiquitous presence of neuro-modulatory signals in the brain that deliver global error signals to neurons.

**Other Comments Or Suggestions:**

L088: 'Pyramidal neurons consist of 70-85% of neurons': I believe this statement is only true for the cortex. The granule cell is presumably the most numerous neuron type in the mammalian brain.

**Other Strengths And Weaknesses:**

A thorough comparison with previously proposed algorithms presented in Table 1 is potentially beneficial for the field, though the authors should make sure that all algorithms are evaluated in a fair condition.

**Questions For Authors:**

Please see the comments above.

**Relation To Broader Scientific Literature:**

I’m not convinced of the novelty of the manuscript. The idea of using apical dendrites for biologically-plausible credit assignments has been discussed previously (J Guerguiev et al., eLife, 2017; J Sacramento et al., NeurIPS 2018; A Payeur et al., Nat Neurosci 2021, …). This work does not appear to offer improvement either in terms of biological plausibility or performance. It is also disappointing that the authors did not discuss any of these works in the related work section.

If $\Theta$ and $W$ are initialized to be the same and $\xi_i$ in Eq. 9 is replaced with $u_i$, the proposed rule becomes equivalent to backprop. Although I am not sure if this particular approximation was implemented before, I’m not surprised by its decent performance, considering previous work explored similar approximations.

**Theoretical Claims:**

Yes, I checked.

In the second line of Eq. 15, the derivatives with respect to i+2, …, n are omitted. Why is that the case?

---

> ### Author Rebuttal · Authors · 2025-03-30
>
> ### 1. Feedback Alignment with MLP on MNIST can achieve 98% test performance.
>
> As stated in Line 325 (left column), to ensure fairness, we adopted the same architecture (784-1024-512-256-10 FC layers, see Appendix C.2) for all algorithms on MNIST.
> The architecture used in the paper you mentioned consists of a 256-256-256-256-256-10 FC structure.
> Differences in network architecture can lead to performance discrepancies, and deeper networks typically achieve higher accuracy.
> For fair comparison, we avoided additional training technologies (like batch normalization, residual connection) as much as possible because potential biases introduced by these additional techniques may have varying effects across different algorithms.
> Therefore, it is not surprising that our reproduced test performance is slightly lower than theirs.
>
> ### 2. The second criterion is problematic considering the neuro-modulatory signals in the brain that deliver global error signals to neurons.
>
> Thanks for your suggestion.
> We are aware of the presence of global signals in the brain, which play an important role in modulating certain mechanisms across neurons.
> However, there is currently no clear biological evidence that these signals represent errors corresponding to the difference between expected and actual outputs, nor that they precisely capture the direction and magnitude required for updating each neuron.
> We agree that further clarification of this criterion is necessary and will add this discussion into Section 2.1.
>
> ### 3. In the second line of Eq. 15, the derivatives with respect to i+2, …, n are omitted
>
> Thank you for pointing this out. There should indeed be an ellipsis when expanding $\mathcal{L}$ in the second line of Eq.15. The rest of Eq. 15 remains correct.
>
> ### 4. Implementation details of previous algorithms are missing.
>
> As stated in Line 325 (left column), to ensure fairness, we use the same model architecture across all learning algorithms for MLPs, CNNs, and RNNs under a certain dataset.
> Detailed model specifications and training configurations can be found in Appendix C.2.
>
> ### 5. The authors did not discuss works using apical dendrites for credit assignments in the related work section.
>
> Thanks for your suggestion.
> While apical dendrites have been discussed in previous literature for credit assignment and have been utilized for various purposes, our study takes advantage of their properties to design and implement more biologically plausible learning algorithms, which differ significantly from existing approaches.
> J Guerguiev et al. (eLife, 2017) primarily aimed to explain how deep learning can be achieved using segregated dendritic compartments, but it did not propose a specific learning algorithm.
> As for Sacramento et al. (NeurIPS 2018), we have acknowledged in Line 200 (right column) that we followed their division strategy for pyramidal neurons.
> A Payeur et al. (Nat Neurosci 2021) investigated burst-dependent synaptic plasticity.
> While their work shares conceptual similarities with ours in terms of apical dendritic processing, the primary objective of their study differs from ours.
> Our proposed method not only satisfies all three criteria but also achieves higher performance compared to existing biologically plausible learning methods.
> That said, we will add the discussion in the related work section. Thank you!
>
> ### 6. If $\Theta$ and $\mathbf{W}$ are initialized to be the same and $ξ_i$ in Eq.9 is replaced with $u_i$, the proposed rule becomes equivalent to backprop. I am not sure if this particular approximation was implemented before.
>
> First, even if $\Theta^T$ and $\mathbf{W}$ are initialized identically, their update directions will differ because $\xi_i = u_i - x_i$ and $x_i$ from $\text{layer}_{i+1}$ cannot be exactly $2u_i$ or $0$ since it comes from lable, so $\xi_i$ and $u_i$ are different in both magnitude and direction. As a result, $\Theta^T$ and $\mathbf{W}$ will update in different directions according to Eq.8 and Eq.9. Furthermore, our formula is derived to minimize the loss, leading to the update formula for $\Theta^T$ in Eq.9. Therefore, replacing $\xi_i$ in Eq.9 with $u_i$ would result in an incorrect update direction. In conclusion, our method is not equivalent to BP.
>
> Second, to the best of our knowledge, there are no existing algorithms that simultaneously satisfy all three criteria while also achieving competitive performance with BP.
> If you could kindly provide the relevant references, we would be pleased to discuss them in our related work section. Thank you!
>
> ### 7. L088: 'Pyramidal neurons consist of 70-85% of neurons': This statement is only true for the cortex.
>
> Thank you for pointing it out. We will clarify this in our revised manuscript.

---

> > ### Comment · Reviewer_Crph · 2025-04-03
> >
> > I thank the authors for their replies. While I appreciate their effort, particularly their meticulous comparisons with some of the previously proposed algorithms, I still believe this work does not offer significant improvement over existing literature either in terms of biological plausibility or performance.
> >
> > The three criteria the authors introduced are neither necessary nor sufficient. While their algorithm is motivated by apical dendrite, it was merely introduced as a computational unit, failing to provide any new biological insights over existing literature on the functional role of apical dendrite for credit assignment (e.g., J Guerguiev et al., eLife, 2017; J Sacramento et al., NeurIPS 2018; A Payeur et al., Nat Neurosci 2021).
> >
> > In terms of performance, there are a series of local learning algorithms that outperform the proposed algorithm. BurstProp (Payeur et al., Nat Neurosci, 2021), SoftHebb (Journe et al., ICLR 2023), and Counter-current learning (Kao & Hariharan, NeurIPS, 2024) achieve 79.9%, 80.3%, and 82.94% on CIFAR-10, respectively, compared to 70.89% performance presented here. Importantly, these works, especially the first two, capture the biological constraints better than the presented work.

---

> > > ### Author Response · Authors · 2025-04-05
> > >
> > > Thank you for your valuable reply!
> > >
> > > ## 1. The three criteria the authors introduced are neither necessary nor sufficient. While their algorithm is motivated by apical dendrite, it was merely introduced as a computational unit, failing to provide any new biological insights over existing literature on the functional role of the apical dendrite for credit assignment.
> > >
> > > The primary objective of this study is not to introduce new biological insights into the functional role of the apical dendrite but rather to demonstrate that a more biologically plausible learning algorithm can be achieved by leveraging the properties of pyramidal neurons, particularly the distinction between apical and basal dendrites.
> > > Our approach enables more effective credit assignment and outperforms existing biologically plausible algorithms. As we mentioned in our 5th response, we will add the necessary discussion in the related work section.
> > >
> > > ## 2. In terms of performance, there are a series of local learning algorithms that outperform the proposed algorithm, like BurstProp (Payeur et al., Nat Neurosci, 2021), SoftHebb (Journe et al., ICLR 2023), and Counter-current learning (Kao & Hariharan, NeurIPS, 2024)
> > >
> > > To ensure that all algorithms were evaluated under consistent conditions, we use the same simple CNN architecture for CIFAR-10 (Line 828), and did not use training techniques such as residual connections or batch normalization.
> > >
> > > BurstProp and SoftHebb can be seen as advancements of STDP and Hebbian learning algorithms, both of which satisfy all three of our proposed criteria.
> > > We aim for our proposed DLL to possess general capabilities comparable to those of backpropagation (BP), including the ability to perform both classification and regression tasks, handle diverse modalities such as images and language, and support multi-layer credit assignment.
> > > For example, in addition to image recognition tasks, our DLL framework can also be applied to train RNNs for regression tasks (Section 4.3), such as next-character prediction and time-series forecasting.
> > > In contrast, methods like BurstProp and SoftHebb may struggle with such tasks, as their designs are not well-suited for recurrent architectures.
> > >
> > > Counter-current Learning violates our third criterion, as it explicitly involves distinct forward and backward phases.
> > > Additionally, their CNN architecture is based on VGG, and the authors employed training techniques such as batch normalization.
> > > We intentionally avoided these techniques to ensure all algorithms were evaluated under consistent conditions.

---

### Official Review · Reviewer_hLAs · 2025-03-13

**Overall Recommendation:** 3

**Summary:**

The article proposes a neural network that is constructed using a local loss and asymmetric weights in the forward and backward passes. The author introduces three criteria for biological plausibility that the neural network should satisfy and demonstrates that the proposed DLL meets these criteria. The author also shows through experiments that DLL outperforms other non-traditional BP networks.

**Claims And Evidence:**

Yes

**Essential References Not Discussed:**

No.

**Experimental Designs Or Analyses:**

The author demonstrates the advantage of DLL over other non-traditional BP methods through experiments, but I think more ablation studies need to be added to prove that the three design criteria for biological plausibility of DLL contribute to improving the model's performance. This will enhance the soundness of this work.

**Methods And Evaluation Criteria:**

Yes

**Other Comments Or Suggestions:**

No.

**Other Strengths And Weaknesses:**

Strengths
    - The article summarizes the characteristics of some non-traditional BP learning methods, extracts three criteria, and designs DLL based on them. Experimental results demonstrate that DLL can outperform previous similar methods.
    - In today's world where SGD is widely used, exploring new learning methods is refreshing and can increase attention to learning approaches within the field.
Weaknesses
    - I think the main issue is that the author should clarify why the three criteria for biological plausibility are important, for example, through ablation studies or theoretical proofs. This would enhance the scientific value of the paper.
    - Although biological plausible local loss and asymmetric weights are used, gradient descent is still employed for training in this paper. Could this be the main reason why DLL outperforms previous work? An important criterion of brain-inspired learning rules is to abandon GD (such as STDP), because the brain does not directly compute gradients.

**Questions For Authors:**

Please see weaknesses.

**Relation To Broader Scientific Literature:**

I think the research in this paper is related to brain-inspired computation and computational neuroscience.

**Theoretical Claims:**

Yes, I checked the math and didn't find any obvious issues.

---

> ### Author Rebuttal · Authors · 2025-03-30
>
> Thank you for your comments and suggestions, which are valuable for enhancing our paper. We are pleased that our contributions are well recognized.
>
> Responses to your concerns and questions are hereby presented:
>
> ### 1. Why are the three criteria for biological plausibility important? More ablation studies are needed to prove that the three criteria contribute to improving the model's performance.
>
> Thanks for your suggestion. The three criteria outlined in our paper were summarized from existing biologically plausible learning algorithms, rather than introduced as a means to enhance model performance.
> Our primary objective is to investigate the biological plausibility of learning algorithms. Based on this investigation and the biological limitations of existing methods, we propose a novel approach that satisfies all three criteria—making it more biologically plausible—while achieving performance comparable to backpropagation.
> We have conducted an ablation study on the backward weight $\Theta$ in Table 3.
> Could you please provide any suggestions on how to design further ablation experiments?
>
> ### 2. Gradient descent is still employed for training. Could this be the main reason why DLL outperforms previous work?
>
>
> Thank you for your insightful comment.
> We agree that STDP is one of the most biologically plausible learning algorithms, as it not only avoids gradient descent but also satisfies the three criteria we summarized.
>
> From the perspective of update rules, the high performance of our method stems from the local neuronal plasticity update rules we derived (Equations 8 and 9), rather than relying on a global gradient backpropagation process.
> We believe that local updates to neuronal plasticity are consistent with the form of computing gradients on local losses, which is a plausible mechanism that could have emerged through long-term evolution.
> Similar approaches have been employed in other biologically plausible learning methods, such as predictive coding, target propagation, and local losses.
> Although gradient descent is not used in STDP, it does not necessarily mean the neuronal plasticity rules can not be derived locally by using gradients.

---

> > ### Comment · Reviewer_hLAs · 2025-04-06
> >
> > The authors have addressed my concerns to some extent. However, given the relatively toy-like structure and the noticeable performance gap, I prefer to keep my original score.

---

> > > ### Author Response · Authors · 2025-04-07
> > >
> > > Thank you very much for your response and for acknowledging that we have addressed your concerns to some extent.
> > >
> > > We sincerely appreciate your feedback and would like to respectfully emphasize that our proposed learning algorithm outperforms existing biologically plausible algorithms that satisfy the three criteria outlined in our study.
> > >
> > > Moreover, similar to backpropagation, our algorithm is general-purpose: it can be applied to both classification and regression tasks, and is suitable for different modalities including language, time series, and vision.
> > >
> > > In contrast, other biologically inspired approaches—such as those based on STDP or Hebbian learning—are typically limited in scope and primarily applicable to image classification tasks.
> > >
> > > Thank you again for your thoughtful feedback.

---

### Official Review · Reviewer_p1AA · 2025-03-13

**Overall Recommendation:** 3

**Summary:**

The paper introduces Dendritic Localized Learning, as an alternative to backpropagation for training neural networks. The goal is to make learning more biologically realistic by addressing three main issues with backpropagation: the requirement of weight symmetry between the forward and backward passes, the use of global error signals, and the separation of learning into distinct forward and backward phases. The authors propose a model inspired by pyramidal neurons, which has separate compartments for different types of information processing. Instead of using the transposed forward weights for the backward pass, DLL introduces a separate set of trainable backward weights. The authors present experiments that show DLL performing better than other biologically plausible learning algorithms while coming close to backpropagation in accuracy. They apply DLL to multilayer perceptrons, convolutional networks, and recurrent networks and compare it to alternatives like feedback alignment and target propagation.

## update after rebuttal

Thanks to the authors for the detailed and thoughtful response. I appreciate the additional experiments on text datasets, the clarification around computational costs, and the effort to demonstrate the stability and scalability of DLL. It’s clear that a lot of work went into addressing the concerns.

That said, my overall view remains the same. While the new experiments help strengthen the paper, the broader limitations — such as the performance gap on complex tasks, the limited scale of evaluation, and the lack of deeper theoretical guarantees — are still there. I still find DLL to be a promising step toward more biologically plausible learning, and the empirical results are encouraging within the scope tested. So, I’m keeping my original score.

**Claims And Evidence:**

The authors claim that DLL satisfies all three criteria for biological plausibility and achieves strong performance on benchmark datasets. The claim that DLL removes the requirement of weight symmetry is supported by the introduction of independent backward weights, which is a reasonable approach. The claim that DLL eliminates the need for a global error signal is also valid because local errors are computed at each neuron. The claim that DLL allows simultaneous forward and backward computation is plausible but could use more experimental validation, especially regarding real-time learning. The claim that DLL achieves performance close to backpropagation is somewhat supported by the reported results, but there is still a noticeable accuracy gap, especially on more complex datasets. There is no strong theoretical guarantee provided for convergence or learning efficiency, which weakens the claim that DLL is a robust learning method.

**Essential References Not Discussed:**

No.

**Experimental Designs Or Analyses:**

The authors conduct a reasonable set of experiments to compare DLL with other learning algorithms, but there are some issues. The choice of datasets is somewhat limited, as all datasets used are relatively small. There is no evaluation of DLL on tasks requiring deeper networks or larger-scale learning. The comparison to backpropagation is primarily based on accuracy, but efficiency is not analyzed. There is no indication of whether DLL requires significantly more training time or memory compared to standard backpropagation. Additionally, there is no ablation study to determine which aspects of DLL are most responsible for its performance. Without such analysis, it is unclear whether the proposed approach is truly necessary or if simpler modifications to existing biologically plausible learning rules could achieve similar results.

**Methods And Evaluation Criteria:**

The proposed method is appropriate for the problem of biologically plausible learning, and the chosen datasets provide a reasonable benchmark. However, the experiments focus mostly on small-scale datasets like MNIST and CIFAR-10. It would be more convincing to see results on more complex datasets like ImageNet or natural language processing tasks. The authors compare DLL against several well-known biologically inspired learning methods, which is a strong aspect of the paper. However, the evaluation mainly considers accuracy, and there is little discussion of computational cost, memory efficiency, or sensitivity to hyperparameters, all of which are important in assessing the practicality of a learning algorithm.

**Other Comments Or Suggestions:**

The authors should include a discussion of DLL’s efficiency and computational cost relative to backpropagation. They should also provide more details on hyperparameter sensitivity and whether DLL requires careful tuning to perform well. The code should be better documented to improve reproducibility. It would also be useful to include a theoretical analysis of DLL’s stability and convergence properties.

**Other Strengths And Weaknesses:**

One of the strongest aspects of this paper is its attempt to provide a biologically inspired learning method that is more plausible than backpropagation while maintaining strong performance. The empirical results suggest that DLL is a promising alternative, and the idea of using separate backward weights is an interesting contribution. However, there are some weaknesses that need to be addressed. The paper lacks a theoretical analysis of DLL’s convergence and stability. The experimental evaluation does not include large-scale tasks, so it is unclear how well DLL scales to more complex problems. There is no discussion of the computational efficiency of DLL compared to backpropagation. The code provided is not well-documented, making it difficult to reproduce the results.

**Questions For Authors:**

How does DLL compare to backpropagation in terms of training time and memory usage? If DLL is slower, what are the trade-offs in terms of biological plausibility versus efficiency?
Would DLL generalize well to large-scale datasets like ImageNet or more complex NLP tasks? Have any experiments been conducted to test its scalability?
Does DLL have any robustness advantages, such as resistance to adversarial attacks or improved performance on noisy data?
Are there specific hyperparameters that need to be fine-tuned for DLL to work well, or is it stable across different settings?

**Relation To Broader Scientific Literature:**

The paper positions DLL as an improvement over previous biologically plausible learning algorithms, particularly feedback alignment and predictive coding. The work is connected to existing neuroscience literature on pyramidal neurons and local synaptic plasticity, which supports its biological inspiration. However, there is little discussion of how DLL relates to other learning methods beyond biologically plausible algorithms. It would be useful to compare DLL’s principles with energy-based models or reinforcement learning approaches that also incorporate local learning rules.

**Theoretical Claims:**

The paper includes mathematical descriptions of DLL, but it does not provide a formal proof of convergence or an analysis of how DLL behaves in different training conditions. While the authors argue that DLL follows biologically plausible principles, they do not establish whether DLL optimizes a well-defined loss function in a way that guarantees stable learning. The absence of such analysis leaves a gap in understanding whether DLL is a reliable alternative to backpropagation or just an interesting theoretical idea with promising empirical results.

---

> ### Author Rebuttal · Authors · 2025-03-30
>
> 1. More experimental validation, especially real-time learning, to show simultaneous forward and backward.
>
> Our approach enables real-time learning, as higher-layer neurons propagate signals backward only when a discrepancy between the output and the label is detected. Otherwise, no adjustments are made, and no backpropagation signals are generated.
> While this mechanism is theoretically sound, we are uncertain about the best way to empirically validate it.
> We would greatly appreciate any suggestions on how to design experiments to test and confirm this behavior.
>
> 2. Accuracy gap on complex datasets.
>
> We primarily focus on comparing the biological plausibility of various learning algorithms rather than optimizing for high performance.
> Within these biological constraints, we propose the DLL algorithm, which further narrows the performance gap with BP.
> As shown in Table 1, DLL achieves the best performance among algorithms that satisfy all three criteria.
>
> 3. Theoretical analysis for convergence.
>
> Please refer to the 4th question of Reviewer 2B1A.
>
> 4. Complex datasets like ImageNet or NLP tasks are ignored.
>
> Given our current computational resources (four 2080 GPUs with 12GB each), we estimate that conducting experiments on the full ImageNet within 7 days would not be feasible.
> Therefore, we chose to evaluate our methods on Tiny-ImageNet.
> The results are presented in our response to the third question from reviewer 2B1A.
>
> For NLP tasks, we have followed [1] to conduct experiments on the "next-character-prediction" task in Section 4.3.
> Additionally, we performed experiments on the text classification datasets Subj and Movie Review (MR), with results summarized below:
> |Method|Subj|MR|
> |-|-|-|
> |BP_TextCNN|88.50%|74.68%|
> |DLL_TextCNN|84.40%|70.79%|
>
> The architecture is the same as the original TextCNN (Kim, EMNLP2014).
>
> 5. Computational cost, memory efficiency, or sensitivity to hyperparameters
>
> The time consumption and memory usage for these experiments are summarized below:
> |Method|Time Consumption (s/epoch)|Memory Usage (MB)|
> |-|-|-|
> |DLL_MLP|44.7|1595.3|
> |DLL_CNN|169.8|1306.9|
> |BP_MLP|31.6|1286.4|
> |BP_CNN|99.0|1272.9|
>
> To fairly compare time consumption across architectures, we used the CPU instead of the GPU.
> DLL requires more training time and memory because both the forward weight $\mathbf{W}$ and backward weight $\Theta$ are updated simultaneously.
> Our design is not driven by computational or memory efficiency; rather, we prioritize biological plausibility.
>
> As for sensitivity to hyperparameters, we have included a comparison of various learning rates and sequence lengths in Figure 3.
>
> 6. The authors do not establish whether DLL optimizes a well-defined loss function in a way that guarantees stable learning
>
> Figure 2(c) shows that models trained with DLL exhibit a stable decrease in training loss.
> This figure is based on time-series forecasting experiments using RNNs on the Electricity dataset.
> Similarly, for MLPs and CNNs, the training losses also show a consistent downward trend.
> We will release all training logs upon acceptance.
>
> 7. No ablation study to determine which aspects of DLL are most responsible for its performance.
>
> Our DLL algorithm is designed based on three criteria that we have summarized from current biologically plausible algorithms.
> It can be viewed as a faster (without iteration) and more biologically plausible implementation of predictive coding (PC), leveraging the unique properties of pyramidal neurons. Our design removes weight symmetry and separates forward and backward computations.
> As a result, the performance of DLL is similar to that of PC under certain conditions, and PC has been shown to achieve performance comparable to backpropagation [1][3].
> We have conducted an ablation study on the backward weight $\Theta$ in Table 3.
> We would appreciate any suggestions on additional ablation studies.
>
> 8. Code lacks documentation. No details about hyperparameter selection.
>
> Upon acceptance, we will release our code with detailed documentation for reproducibility.
> The selection of hyperparameters follows the similar choices made in [1][2].
>
> 9. Discussion of other learning methods.
>
> We will add related papers on apical dendrites.
> Please refer to the 5th question of reviewer Crph.
>
> 10. Robustness and scalability?
>
> Robustness is an interesting aspect, and we plan to explore it in future work.
>
> We conduct scalability experiments:
>
> |DLL-MLP|MNIST|
> |-|-|
> |784-1024-10|71.15%|
> |784-1024-512-10|89.61%|
> |784-1024-512-256-10|97.57%|
>
> All MLPs are trained fairly, and the results show the scalability of DLL.
>
> [1] Millidge B, et al. Predictive coding approximates backprop along arbitrary computation graphs. Neural Computation, 2022.
>
> [2] Bartunov S, et al. Assessing the scalability of biologically-motivated deep learning algorithms and architectures. NeurIPS, 2018.
>
> [3] Salvatori T, et al. A Stable, Fast, and Fully Automatic Learning Algorithm for Predictive Coding Networks. ICLR, 2024.

---

### Official Review · Reviewer_2B1A · 2025-03-14

**Overall Recommendation:** 3

**Summary:**

The paper introduces Dendritic Localized Learning (DLL), a biologically plausible learning algorithm inspired by the structure and plasticity of pyramidal neurons. The motivation behind DLL is to address three fundamental biological limitations of backpropagation:

- Weight symmetry – Backprop requires symmetric forward and backward weights, which is not biologically plausible.
- Global error signals – Biological neurons primarily use local learning rules rather than propagating global error through all the layers.
- Dual-phase learning – Backprop separates forward and backward passes, whereas biological learning does not have such a strict separation.

The proposed DLL framework models neurons with three compartments (soma, apical dendrite, basal dendrite) and introduces trainable backward weights  instead of using transposed forward weights in error propagation. The algorithm enables local error computation within neurons and allows for simultaneous weight updates, satisfying biological plausibility criteria.

## update after rebuttal
I appreciate the authors effort in rebuttal. While I understand the importance of maintaining uniformity across algorithms, I do believe that that for a new learning algorithm, it is also useful to measure or discuss the compatibility with the standard regularization techiniques like weight decay, batch normalization. Also, I don't fully agree that adding dropout for instance makes the approach biologically implausible. Noise is a salient feature of the learning machinery of the brain and dropout, adds a source of noise.

I am not sure if it is a typo, but the authors report DLL to be outperforming on the most challenging dataset (TinyImageNet) which seems highly unlikely and the numbers are quite high compared to Cifar10 and Cifar100. It is also a bit concerning to see model performing better on TinyImageNet compared to Cifar10.

With these concerns remaining, I will retain my original score and do not feel confident to champion the paper for acceptance.

**Claims And Evidence:**

The authors claim that DLL satisfies all three criteria for biological plausibility while maintaining strong empirical performance compared to prior biologically plausible learning algorithms. They provide:
- Mathematical derivation of the DLL learning rule and its update equations.
- Empirical performance on MLPs, CNNs, and RNNs across various datasets.
- Comparisons with other biologically plausible algorithms, showing that DLL achieves superior accuracy among methods that meet all three criteria.

**Essential References Not Discussed:**

Not to reviewers' knowledge.

**Experimental Designs Or Analyses:**

Overall, the experimental design in sound. I have a few concerns:
- The performance of backpropagation on cifar10 with CNN (75%) seems quite low. Can the authors provide any justification for this,
- Authors do not discuss the effect of common regularizations like weight decay or batch normalization. Does it work for modern architectures like ResNets?
- It would be insightful to see how DLL performs on more complex datasets like Cifar100 or TinyImageNet.

**Methods And Evaluation Criteria:**

The experimental methodology is well-structured and covers different architectures (MLPs, CNN and RNNs) and datasets. This showcases the versatility and applicability of their learning algorithm.

**Other Comments Or Suggestions:**

In addition to addressing the concerns raised above, manuscript would benefit from

- Discussion on convergence properties and guarantees would make the manuscript much stronger.
- Discussion on how the authors believe the performance gap between backpropagation and DLL can be bridged and its applicability to modern architectures and complex datasets.

**Other Strengths And Weaknesses:**

- DLL provides a promising approach which fulfills biologically plausibility criterion and provides comparable performance.
- Well written and structured and easy to follow

For weaknesses, see the concerns mentioned above.

**Questions For Authors:**

Q1) Can the author comment on the applicability of DLL on deeper CNNs and modern architectures like ResNets.

Q2) Can the authors explain the low performance with backpropagation on  CIfar10 with CNN. And the effect of weight decay and batch normalization on DLL performance and convergence.

**Relation To Broader Scientific Literature:**

The paper clearly situates DLL within the field of biologically plausible learning.

**Theoretical Claims:**

The paper presents a mathematical derivation of the DLL learning rule and its update equations. Reviewer did not verify the correctness of their derivation.

---

> ### Author Rebuttal · Authors · 2025-03-30
>
> ### 1. Performance of BP on CIFAR10 with CNN (75%) is low.
>
> Firstly, as stated in Line 325 (left column), to ensure fair comparisons between different algorithms, we used the same architecture for all methods on a given dataset.
> Specifically, for CIFAR-10, we employed a CNN with three convolutional layers (see Appendix C.2.2) without incorporating deep learning techniques such as batch normalization, residual connections, or dropout.
> This design choice was made to maintain full biological plausibility and prevent potential biases that could arise from the selective application of these techniques across different algorithms.
> Under these conditions, the accuracy achieved by our BP-CNN on CIFAR-10 is reasonable.
>
> Secondly, previous studies like [1][2][3], which also did not incorporate additional training techniques, reported test accuracies of only achieving 60%-70% on CIFAR10 using similar CNN architectures.
> Our primary objective was to establish a fair and unbiased evaluation framework, ensuring that comparisons reflect the intrinsic differences between algorithms rather than the influence of external training enhancements.
>
> ### 2. Effect of common regularizations like weight decay or batch normalization. Does DLL work for modern architectures like ResNets?
>
> As mentioned in the last question, we did not incorporate additional training techniques.
> If we were to use them, special design considerations would be necessary, as they may violate certain criteria.
> For example, batch normalization and dropout behave differently during training and testing, and directly applying them to our DLL framework would break Criterion 3 (non-two-stage training).
> We acknowledge that incorporating such techniques, particularly ResNets, could be beneficial for future optimizations and will consider them in subsequent work.
> However, our primary objective in this paper was to explore and evaluate the biological feasibility of the algorithms without relying on external enhancements.
>
> ### 3. How does DLL perform on more complex datasets like CIFAR100 or Tiny-ImageNet?
> We followed your suggestions and trained CNNs with BP and DLL on CIFAR-100 and TinyImageNet.
> Consistent with previous work [1], we report test accuracy for CIFAR-100 and test error rate for TinyImageNet.
>
> |Method|CIFAR100 (test accuracy)|TinyImageNet (test error rate)|
> |-|-|-|
> |BP_CNN|44.5%|78.6%|
> |DLL_CNN|38.6%|82.9%|
>
> Note that we do not use any additional training techniques such as batch normalization or residual connections. Our CNN architecture is similar to that in [1].
> For CIFAR-100, the CNN consists of four convolutional layers with channel configurations of 3-64-64-128-64, followed by two fully connected layers.
> For TinyImageNet, the CNN consists of five convolutional layers with filter configurations of 3-64-64-128-128-64, followed by two fully connected layers.
>
> ### 4. Discussion on convergence properties and guarantees would make the manuscript much stronger.
>
> Thanks for your advice. We will add the following discussion on convergence properties and guarantees:
> The loss function (Eq. 3) is designed to minimize the discrepancy between the top-down predictions and bottom-up outputs of each pyramidal neuron in the network.
> To achieve this, we employ local gradient descent–based learning rules and neural plasticity mechanisms to update both forward and backward weights.
> During each iteration, the differences between the network’s predictions and the ground truth propagate back through localized errors, effectively coordinating all neurons in an orchestrated manner.
> As a result, neural responses collectively refine predictions over successive iterations, gradually reducing local errors and driving the network toward convergence.
> While providing formal convergence proofs remains challenging due to the network’s nonlinear operations, our empirical results consistently demonstrate a steady decrease in loss throughout training, supporting the stability and effectiveness of our approach.
>
> ### 5. How can the performance gap between BP and DLL be bridged?
>
> Currently, most biologically plausible learning algorithms exhibit a significant performance gap compared to BP. In this study, our primary goal is to move closer to BP while preserving biological plausibility.
> If the sole objective were performance improvement, specialized training techniques such as normalization and dropout—adapted to our proposed DLL algorithm—could be explored in future work.
>
> Reference
>
> [1] Bartunov S, Santoro A, Richards B, et al. Assessing the scalability of biologically-motivated deep learning algorithms and architectures. Advances in neural information processing systems, 2018, 31.
>
> [2] Millidge B, Tschantz A, Buckley C L. Predictive coding approximates backprop along arbitrary computation graphs. Neural Computation, 2022, 34(6): 1329-1368.
>
> [3] Salvatori T, Song Y, Yordanov Y, et al. A Stable, Fast, and Fully Automatic Learning Algorithm for Predictive Coding Networks. ICLR, 2024

---

### Official Review · Reviewer_7p2D · 2025-03-16

**Overall Recommendation:** 2

**Summary:**

The paper focuses on biological plausibility. Looking at prior work, this paper proposes three different metrics for biological plausibility including (i) asymmetry between the forward and the backward weights, (ii) local losses, and (iii) non two-stage training. With that, the paper proposes a new learning system i.e., Dendritic Localized Learning (DDL), which they compare against other local learning approaches as well as end-to-end backpropoagation. The obtained results on computer-vision classification datasets, time-series forecasting datasets, and NLP datasets highlights that potential of DDL in comparison to other local learning approaches.

**Claims And Evidence:**

While the general claims are well-supported, the claim regarding the three properties being sufficient is really weak without any citation. I attempted to elaborate on that more in the later sections.

**Essential References Not Discussed:**

Yes, quite a lot of them. The paper didn't cover any of the most recent methods in this space, with almost all comparative analysis limited to papers from the late 2010s. E.g., local losses itself is a very active space, with LoCo being one of the most prominent papers in this space: https://arxiv.org/abs/2008.01342). There are now many extensions of this. I also elaborate on this in my other comments.

**Experimental Designs Or Analyses:**

Yes. The evaluation and analysis is straightforward as the paper focuses on standard benchmark datasets.

**Methods And Evaluation Criteria:**

Yes

**Other Comments Or Suggestions:**

- Line 263 (left column): mention that 'f' represents the non-linearity (which is clarified in Algorithm 1)
- Line 318 (left column): "correctly backpropagated" -> "correct predicted"

**Other Strengths And Weaknesses:**

# Strengths
- Well motivated problem
- Simple and well-motivated method
- Well-written paper
- Comprehensive coverage in terms of evaluation
# Weakness
- Poorly supported claim of just three essential properties without any citation from the neuroscience literature that argues about the biological plausibility of these ideas. They seemed to emerge out of the blue. There are many other criterions explored in prior work. E.g., https://arxiv.org/abs/2302.01647 argued that self-supervised learning and no dependence within layers are also essential for biological plausibility. Hence, the contribution of estimating essential properties based on prior work is not the first. Furthermore, it is hard to claim that the list is complete with these three properties.
- Ignores almost all of the recent related work in this space. Particularly, there has been a lot of focus on recent methods that attempt to make local learning methods scale to large-scale settings (such as ImageNet), while the papers cited in the current work even fail to work on MNIST at times. With correct local learning methods that detach one layer from another, they do satisfy C3 naturally as there is no end-to-end update.
- Weak architecture selection that already results in poor performance as a start, even with end-to-end BP. E.g., in table 1, the performance of CIFAR-10 with a CNN is just 75%, which is really weak.

**Questions For Authors:**

- How were the architectures selected? Why is the performance of the selected architectures poor on CIFAR-10?
- How were the hyper parameters tuned? Were they the same for all methods?
- Can this method be adapted to residual networks?

**Relation To Broader Scientific Literature:**

The claim of a new contribution based on the analysis of properties of local learning system ignores other work in this space. This is not a unique contribution of the paper as previous papers did the same. Furthermore, the three learning criterions covered in the work can't be considered complete (e.g., see https://arxiv.org/abs/2302.01647 -- I'll cover this more in the weaknesses section). Finally, the paper ignored all of the recent local learning methods, and only focused on old ones. Particularly, the paper ignored all methods that attempt to scale to larger datasets (e.g., see https://arxiv.org/abs/2008.01342 and https://arxiv.org/abs/2302.01647). Local losses are a very active area of research. The covered literature on this is very weak with just one paper.

**Theoretical Claims:**

N/A

---

> ### Author Rebuttal · Authors · 2025-03-30
>
> ### 1. Poorly supported claim of 3 criteria without any citation. There are many other criteria explored in prior work.
>
> Firstly, in Section 2.2, we provide a detailed introduction and evaluation of all representative biologically plausible learning algorithms. These algorithms served as the basis for the three criteria, as mentioned in Line 33 (left column). However, we acknowledge that we should have included more explicit citations before discussing these criteria in Section 2.1, and we appreciate the reviewer’s suggestion in this regard.
>
> Secondly, our work specifically focuses on biologically plausible supervised learning algorithms. For instance, when discussing Hebbian Learning and STDP, we highlighted their limitation in leveraging supervised learning signals (Line 188, left column). While we acknowledge that studies such as [4][5] have proposed alternative criteria—such as self-supervised learning and independence between layers—we emphasize that these approaches are often based on unsupervised learning. We fully recognize the value of these criteria in assessing biological plausibility; however, it is challenging to incorporate all possible criteria within a single paper.
>
> To address this, we will add a discussion and cite relevant studies in our revised manuscript to clarify our motivation. We appreciate the reviewer’s insightful feedback.
>
> ### 2. Ignorance of recent related work [3][4] on local learning. Local learning methods satisfy C3 naturally as there is no end-to-end update.
>
> Firstly, as mentioned in the last question, our work focuses on supervised learning algorithms.
> [3] proposed a layer-wise self-supervised pre-training algorithm based on random masking and image recovery, while [4] explored deepening model layers within unsupervised contrastive learning frameworks.
> Although these methods help scale local learning approaches to larger settings, their biological plausibility is limited, as they do not fully adhere to Criteria 1 and 3.
> In this study, our primary goal is to take a step closer to BP while maintaining biological plausibility.
> To achieve this, we designed the DLL algorithm based on three key criteria derived from existing biologically plausible learning algorithms.
> That said, we acknowledge the relevance of [3] and [4] and will incorporate a discussion of these works in Section 2.2 and the related work section.
>
> Secondly, we define Criterion 3 as non-two-stage training, meaning there is no strict temporal segregation between forward and backward processes.
> We believe that not all local learning methods satisfy this criterion. For instance, difference target propagation is a local learning method but does not meet Criterion 3, as we discussed in Line 143 (right column).
> While the training methods in [3] and [4] do not involve end-to-end updates, they still exhibit a clear separation between forward and backward processes.
>
> ### 3. Writing suggestions: Line 263 and Line 318.
> Thanks for pointing them out. We will correct them in our revised manuscript.
>
> ### 4. Why is the performance of the selected architectures poor on CIFAR-10?
>
> Please refer to the first question of reviewer 2B1A.
>
> ### 5. How were the architectures selected? How were the hyperparameters tuned? Were they the same for all methods?
>
> Yes, as stated in Line 325 (left column) and Appendix C.2, we ensured that all algorithms were evaluated using the same hyperparameters and architectures for a given dataset.
> Our choices for architecture and hyperparameter settings were based on the methodologies outlined in [1][2].
>
> ### 6. Can this method be adapted to residual networks?
>
> We recognize that there is biological evidence supporting inter-regional and inter-layer neuronal connections.
> However, whether these connections function equivalently to residual connections in deep learning remains an open question.
> Residual connections were originally introduced to mitigate the gradient vanishing problem in deep learning models, which is less of a concern for local learning-based approaches like ours.
> As a result, their potential benefits may not be as relevant or impactful in our setting.
>
> We acknowledge the importance of exploring such architectural enhancements.
> In future work, we will consider incorporating residual connections to further investigate their potential impact and applicability.
>
> ### Reference
>
> [1] Bartunov S, Santoro A, Richards B, et al. Assessing the scalability of biologically-motivated deep learning algorithms and architectures. NeurIPS 2018.
>
> [2] Millidge B, Tschantz A, Buckley C L. Predictive coding approximates backprop along arbitrary computation graphs. Neural Computation, 2022, 34(6): 1329-1368.
>
> [3] Siddiqui S, Krueger D, LeCun Y, et al. Blockwise Self-Supervised Learning at Scale. TMLP.
>
> [4] Xiong Y, Ren M, Urtasun R. Loco: Local contrastive representation learning. NeurIPS, 2020

---

### Decision · Program_Chairs · 2025-05-01

**Decision:**

Accept (poster)

**Comment:**

Summarizing all the reviews, my interpretation and summary is the following:
-- Solid work. No technical flaws. Authors addressed (most of) the questions about benchmarks and other datasets.
-- Novelty has been questioned. There is a lot of work on dendrites and local learning. Some of this work was not adequately cited in the original submission and the authors acknowledged this and will cite the work.
-- There was discussion about whether the 3 key criteria outlined by the authors are sufficient and the consensus is that they are important criteria but not sufficient for biological plausibility.
-- There are questions about computation time and extension to more complex problems.
There were mixed opinions among reviewers about the overall contribution.